# Nanoparticulate cell-free DNA scavenger for treating inflammatory bone loss in periodontitis

Hanyao Huang [1,2,3,14], Weiyi Pan[1,4,12,14], Yifan Wang [5,13,14], Hye Sung Kim [2,6], Dan Shao [2,7], Baoding Huang[2,8], Tzu-Chieh Ho [2], Yeh-Hsing Lao[2], Chai Hoon Quek[2], Jiayu Shi[9], Qianming Chen[1,4,12], Bing Shi[1,3], Shengmin Zhang[5] ✉, Lei Zhao [1,10] ✉ & Kam W. Leong [2,11] ✉

Periodontitis is a common type of inflammatory bone loss and a risk factor for systemic diseases. The pathogenesis of periodontitis involves inflammatory dysregulation, which represents a target for new therapeutic strategies to treat periodontitis. After establishing the correlation of cell-free DNA (cfDNA) level with periodontitis in patient samples, we test the hypothesis that the cfDNA-scavenging approach will benefit periodontitis treatment. We create a nanoparticulate cfDNA scavenger specific for periodontitis by coating selenium-doped hydroxyapatite nanoparticles (SeHANs) with cationic polyamidoamine dendrimers (PAMAM-G3), namely G3@SeHANs, and compare the activities of G3@SeHANs with those of soluble PAMAM-G3 polymer. Both G3@SeHANs and PAMAM-G3 inhibit periodontitis-related proinflammation in vitro by scavenging cfDNA and alleviate inflammatory bone loss in a mouse model of ligature-induced periodontitis. G3@SeHANs also regulate the mononuclear phagocyte system in a periodontitis environment, promoting the M2 over the M1 macrophage phenotype. G3@SeHANs show greater therapeutic effects than PAMAM-G3 in reducing proinflammation and alveolar bone loss in vivo. Our findings demonstrate the importance of cfDNA in periodontitis and the potential for using hydroxyapatite-based nanoparticulate cfDNA scavengers to ameliorate periodontitis.

Periodontitis is a common chronic disorder that involves oral microbe-related inflammatory bone loss and local destruction of periodontal tissue, and is associated with systemic diseases such as diabetes and rheumatoid arthritis[1–3]. In periodontitis, microbial inflammatory insults, and abnormal host-microbiota interactions inappropriately and continuously activate the immune system and induce a proinflammatory immune response, leading to inflammatory bone loss[4,5]. Proinflammatory immune response is initiated by toll-like receptors (TLRs) and other pattern-recognition receptors that detect and respond to pathogen-, damage-, and microbe-associated molecular patterns (PAMPs, DAMPs, and MAMPs)[6]. Inappropriate immune

system activation of TLR9 is a major factor in many inflammatory diseases[7]. In patients with periodontitis, TLR9-positive cells can be found in the basement membranes and the suprabasal layers of the oral epithelium and extend to all layers of the pocket epithelium[8]. Destruction of periodontal tissue is also associated with increased TLR9 levels[9]. Animal studies demonstrate that TLR9-mediated immune response can induce inflammatory periodontal bone loss, and TLR9-deficient mice are resistant to periodontitis[10,11]. In vitro studies also indicate that TLR9 modulates the immune response against periodontal pathogens[10,11]. It is therefore not unreasonable to speculate that TLR9 plays an important role in the pathogenesis of periodontitis.

TLR9 preferentially recognizes DNA sequences containing unmethylated cytosine-phosphate-guanosine (CpG) oligodeoxynucleotides, which are highly frequent motifs in bacterial DNA (bDNA), and triggers proinflammatory cytokine response[12]. Cell-free DNA (cfDNA), including endogenous nuclear and mitochondrial DNA released by damaged host cells, and exogenous bacterial or viral DNA[13,14], serve as ligands to TLR9. In patients with periodontitis, cfDNA level in the gingival crevicular fluid is correlated with the degree of periodontitis[15–17]. In vitro studies also demonstrate that bDNA, as part of cfDNA, can induce TLR9-mediated inflammation related to periodontitis[18–20]. Moreover, cfDNA is implicated in the chronic inflammatory bone loss of rheumatoid arthritis[21,22]. Although the linkage between cfDNA and inflammatory bone loss in periodontitis has not been well established, we hypothesize that targeting cfDNA might be a viable therapeutic strategy to reduce bone loss in periodontitis.

We and others have previously shown that cationic nanomaterials can have a therapeutic effect on many inflammatory diseases in animal models by scavenging the negatively charged cfDNA[23], which include acute liver failure[24], lupus[25], cancer metastasis[26], influenza infection[27], sepsis[28], psoriasis[29], and inflammatory bowel disease[30]. After establishing the correlation between cfDNA in saliva and serum of patients with periodontitis, we designed a nanoparticulate scavenger specifically for this application. We coated selenium-doped hydroxyapatite nanoparticles (SeHANs) with the cationic dendrimer PAMAM-G3 to form the cationic nanoparticle, G3@SeHANs. SeHAN was chosen because of the good compatibility of hydroxyapatite in bone tissue[31,32] and the osteogenic and antioxidant properties of selenium[33–36]. A cationic nanoparticle was preferred over a soluble polycation because of its potentially higher TLR9-inhibitory activity[37], and naturally amenable to nanoparticle targeting.

In this study, we investigated the relationship between cfDNA levels and severity of periodontal inflammation, characterized the DNA binding affinity and TLR9-inhibitory activity of G3@SeHANs, elucidated the cfDNA-scavenging mechanisms of G3@SeHANs and PAMAM-G3 and their activities in modulating the M1/M2 macrophage phenotypes, and finally compared the efficacy of these two scavenging configurations in reducing inflammatory bone loss in a ligature-induced periodontitis murine model. Taken together, the results validated the hypothesis that a nanoparticulate DNA scavenger would be valuable in treating inflammatory bone loss in periodontitis.

## Results

### cfDNA levels in saliva and serum are correlated with the severity of periodontal inflammation

The periodontal examination applied to each patient (patient demographics shown in Supplementary Table 1) included dental plaque index (PLI), bleeding index (BI), and probing depth (PD), which could directly show the severity of periodontal inflammation and reflect the inflammatory bone loss of the patients; the periodontal status of each patient was in turn characterized as healthy, gingivitis, and periodontitis (Supplementary Fig. 1). Gingivitis was a mild and early form of periodontal inflammation, which could make the gum red and swollen. Gingivitis was reversible, but without control, it could also turn into periodontitis with irretrievable bone loss.

cfDNA levels in saliva and serum were significantly higher in periodontitis patients than in gingivitis and healthy volunteers, and no significant difference was found between gingivitis and healthy volunteers (Fig. 1a). The correlation of cfDNA levels in saliva and serum with the progression of periodontitis in terms of PLI, BI, and PD was shown in Fig. 1b and Supplementary Figs. 2–4. Besides the respective average and maximum parameters, cfDNA levels in saliva showed the strongest correlation with periodontitis (PLI > 2%, BI > 2%, and PD > 4 mm%), and the correlations were much stronger in saliva than in serum (Fig. 1c).

### Cationic G3@SeHANs can function as a DNA scavenger

Based on the aforementioned correlation of cfDNA and periodontitis, and the participation of TLR9 activation in the periodontitis immune response[10], we synthesized nanoparticulate cfDNA scavenger, namely G3@SeHANs by coating the cationic dendrimer PAMAM-G3 on the surface of SeHANs (Fig. 2a). The synthetic SeHANs showed a hierarchical structure similar to natural bone apatite on scanning electron microscopy (SEM) (Supplementary Fig. 5); Fourier-transform infrared spectroscopy (FTIR) and X-ray diffraction (XRD) demonstrated a high similarity of chemical composition and phase characteristics between SeHANs and natural bone apatite (Supplementary Figs. 6, 7). G3@SeHANs were positively charged and better dispersed than bare SeHANs based on SEM and transmission electron microscopy (TEM) (Fig. 2a and Supplementary Fig. 8). PAMAM-G3 coating of the SeHANs was confirmed by elemental mapping in TEM (Supplementary Fig. 8), X-ray photoelectronic spectroscopy (XPS) (Fig. 2b–d), FTIR (Fig. 2e), thermogravimetric (TG) analysis (Fig. 2f), zeta potential (Fig. 2g), and Brunauer-Emmett-Teller (BET) surface area analysis (Fig. 2h). The size of the particles increased slightly following coating (Fig. 2i), but the crystallinity did not change (Fig. 2j) − the particles remained insoluble in the aqueous medium. PAMAM-G3 grafting reduced the selenium release rate of SeHANs (Supplementary Fig. 9a), indicating a slower biodegradation rate. The lower selenium content of G3@SeHANs compared with bare SeHANs (Supplementary Fig. 9b) was likely due to selenium release during grafting. More importantly, PAMAM-G3 grafting endowed G3@SeHANs with greatly enhanced DNA binding affinity in comparison with bare SeHANs (Supplementary Fig. 10a), and the grafting reduced the potential cytotoxicity of soluble PAMAM-G3 (Supplementary Fig. 10b).

We then examined whether G3@SeHANs could inhibit TLR activation to confirm that the cationic polymer coating empowered the nanoparticle with immune-modulatory effects, by using HEK-TLR3, -TLR4, -TLR8, and -TLR9 reporter cells and corresponding TLR agonists: poly(I:C) dsRNA, lipopolysaccharide (LPS), ORN06 ssRNA, and CpG DNA, respectively (Supplementary Figs. 11, 12). Both G3@SeHANs and bare SeHANs inhibited CpG-induced TLR9 activation in a dose-dependent manner, but G3@SeHANs caused greater inhibition than bare SeHANs; 10 μg/mL of G3@SeHANs caused a similar level of inhibition as 2 μg/mL of soluble PAMAM-G3 (Supplementary Fig. 11a, b). The G3@SeHANs also inhibited poly(I:C)-induced TLR3 activation, LPS-induced TLR4 activation, and ORN06-induced TLR8 activation to a greater extent than bare SeHANs (Supplementary Fig. 12). Together, G3@SeHANs exhibited a greater ability to scavenge anionic TLR agonists, including DNA, and inhibit TLR activation than bare SeHANs, indicating the superiority of the nanoparticulate configuration of the scavenger over its soluble counterpart.

### Cellular TLR9 signaling can be inhibited by scavenging periodontitis-derived cfDNA

Saliva and serum from periodontitis patients induced significantly higher TLR9 activation in HEK-Blue TLR9 cells than that from healthy donors (Fig. 3a, b). Such TLR9 activation was inhibited by both PAMAM-G3 and G3@SeHANs (Fig. 3c, d). We then used two common types of gingival cells−human gingival keratinocytes (HGKs) and human gingival fibroblasts (HGFs)−to collect DAMPs. DAMPs from both cells and mitochondria (mtDAMPs) caused significant increases in TLR9 activation (Fig. 3e), and these increases were mitigated by adding G3@SeHANs and PAMAM-G3. MAMPs collected from two common periodontitis-related bacteria, *Porphyromonas gingivalis* and *Fusobacterium nucleatum* also caused a significant increase in TLR9 activation; *Porphyromonas gingivalis*-derived MAMPs caused a significant increase in TLR9 activation, which was reduced by adding

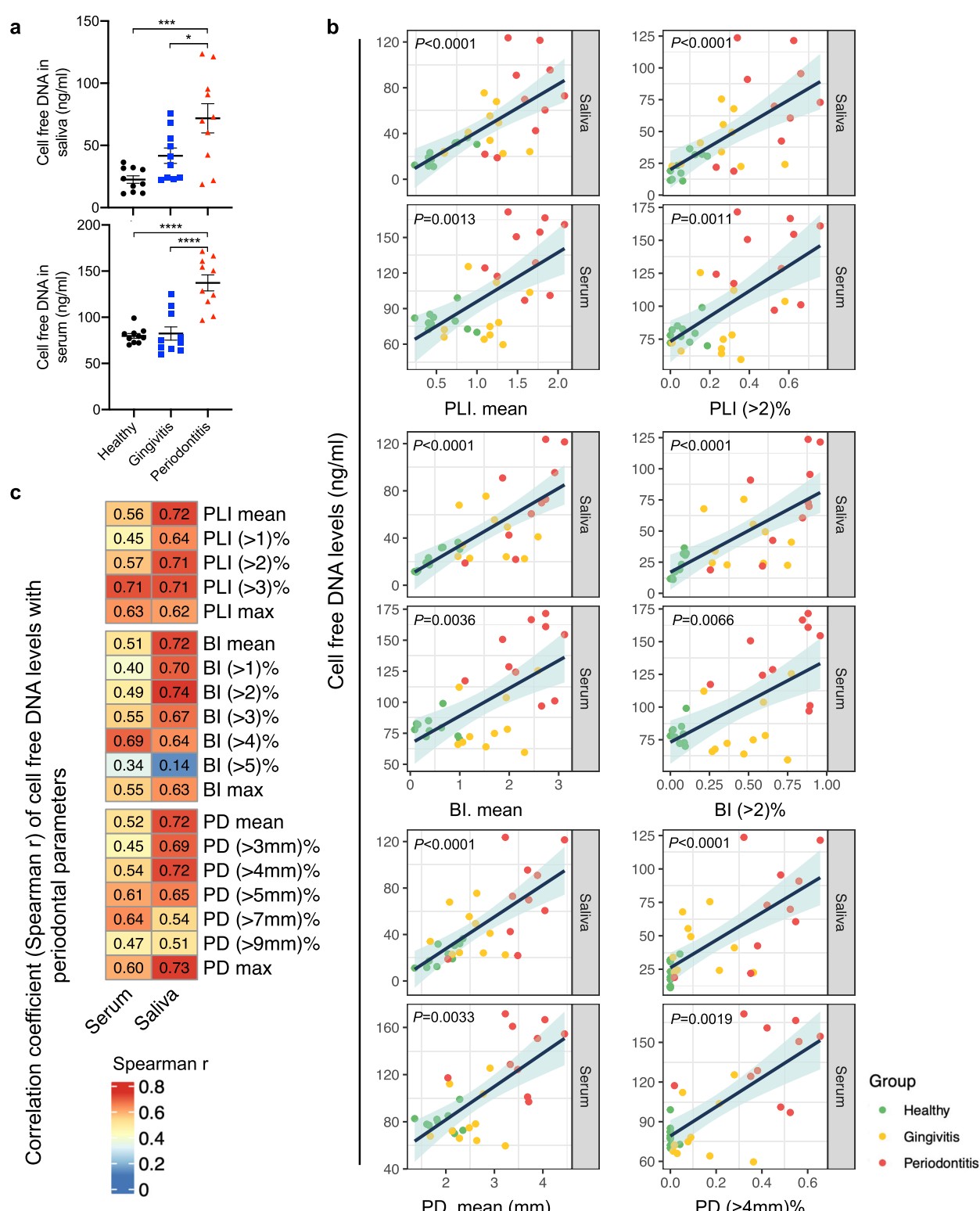

**Fig. 1 | cfDNA concentration in saliva and serum is positively correlated with the severity of periodontal inflammation, and cfDNA in saliva showed stronger correlation. a** cfDNA levels in saliva and serum of healthy volunteers ($n = 10$), gingivitis patients ($n = 10$) and periodontitis patients ($n = 10$). Data are means ± SEM; differences were assessed by one-way analysis of variance and Tukey's multiple comparisons test. *$P < 0.05$, **$P < 0.01$, ***$P < 0.001$, ****$P < 0.0001$ (Saliva, $P = 0.0004$ between Healthy and Periodontitis, $P = 0.0302$ between Gingivitis and Periodontitis; Serum, $P < 0.0001$ between Healthy and Periodontitis, $P < 0.0001$ between Gingivitis and Periodontitis). **b** Scatter plot of cfDNA levels and

parameters of periodontal examination. The color of the dot represents the group of subjects. The dark blue line is the fitted regression line, and the light blue shading around it is the 95% confidence interval. **c** Heatmap of the Spearman's correlation coefficient "r" between cfDNA levels in body fluids and parameters of periodontal examination. PLI: plaque index (1-3); BI: bleeding index (1-5); PD: probing depth (mm). "Mean" value is the arithmetic mean of the records of all examined sites. "(>n)%" is the percentage of site with parameter greater than "n" to the total number of sites. "Max" is the maximum of all sites.

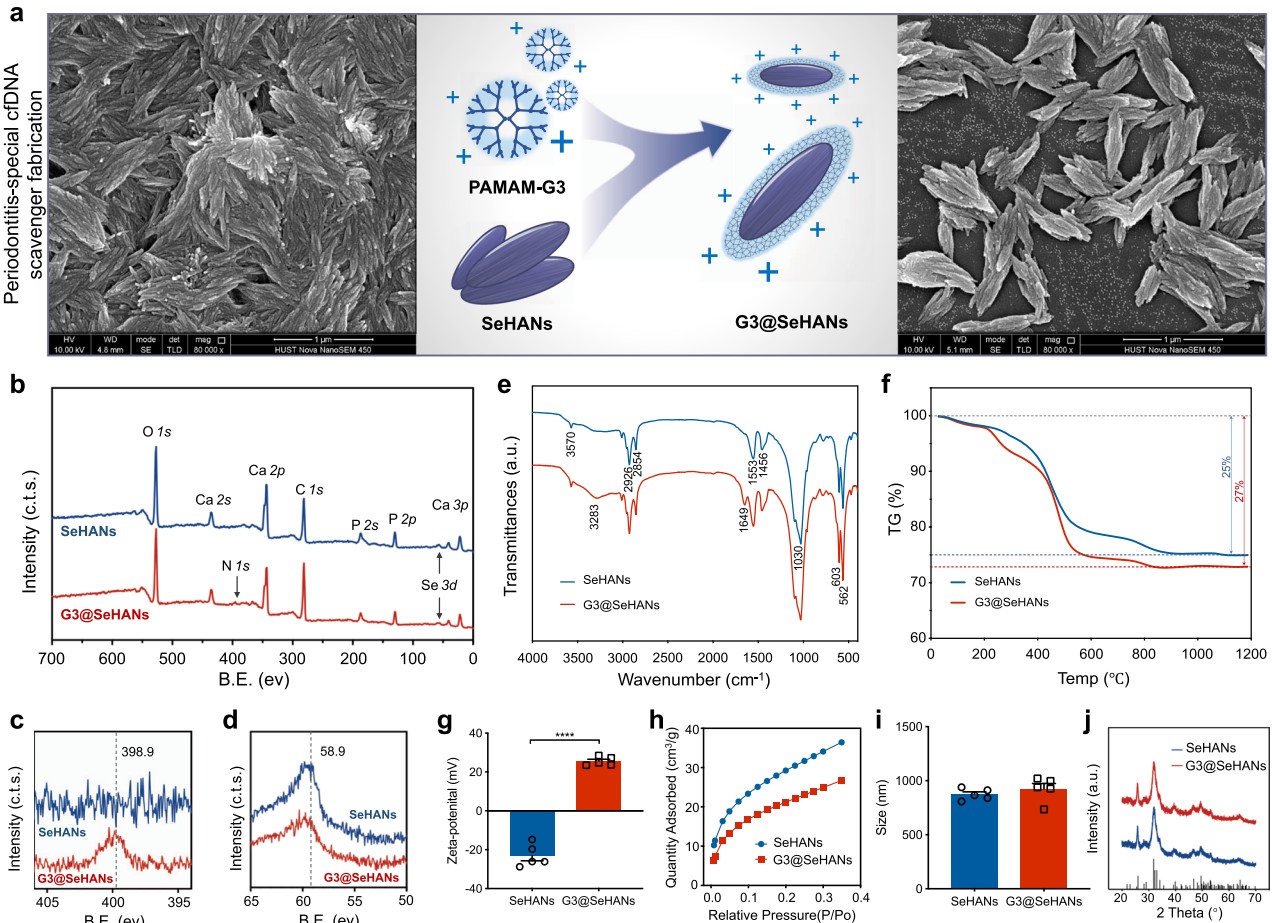

**Fig. 2 | Characterization of G3@SeHANs. a** SEM imaging of SeHANs before and after coating with PAMAM-G3. **b–d** XPS spectra. The *N 1s* photoelectron peak (398.9 eV) is present only in G3@SeHANs; Se *3d* peaks (58.9 eV) are present in both SeHANs and G3@SeHANs. Representative images were from 3 independent experiments. **e** FTIR spectra. Both SeHANs and G3@SeHANs exhibit characteristic phosphate bands at 562, 603, and 1030 cm⁻¹, and bands corresponding to O-Se-O bending vibration (783 cm⁻¹) and hydroxyl groups (3570 cm⁻¹). Peaks due to amino stretching vibration (3283 cm⁻¹) and bending vibration of the amide N-H bond (1649 cm⁻¹) are present only in G3@SeHANs. **f** The thermal decomposition rate of the nanoparticles changes after coating with PAMAM-G3. **g** Zeta potential of SeHANs and G3@SeHANs. Data are means ± SEM; differences were assessed by two-tailed Student's *t*-test (*n* = 5) (*P* < 0.0001). **h** BET analysis of SeHANs and G3@SeHANs. **i** Size measurement of SeHANs and G3@SeHANs. Data are means ± SEM; differences were assessed by two-tailed Student's *t*-test (*n* = 5) (*P* = 0.3898). **j** XRD analysis of SeHANs and G3@SeHANs. *\*P* < 0.05, *\*\*P* < 0.01, *\*\*\*P* < 0.001, *\*\*\*\*P* < 0.0001.

G3@SeHANs and PAMAM-G3, while the increase caused by *Fusobacterium nucleatum*-derived MAMPs was only reduced by G3@SeHANs (Fig. 3e). To investigate the type of periodontitis-derived cfDNA responsible for activating the TLR9, we used cfDNA isolated from periodontitis saliva and serum, genomic DNA (gDNA) and mitochondrial DNA (mtDNA) as part of DAMPs, and CpG DNA as part of MAMPs to stimulate the HEK-TLR9 receptor cells. We found that cfDNA from periodontitis saliva but not serum, mtDNA but not gDNA, and CpG DNA induced TLR9 activation, and consistently G3@SeHANs reduced all these TLR9 activations, while the soluble PAMAM-G3 only reduced the activation by mtDNA and CpG DNA (Fig. 3f). Notably, DNA extracted from saliva and serum did not show the same activation capacity on TLR9 reporter cells, which might be due to their different origins. The size of DNA fragment characteristics of the extracted cfDNA also showed that the serum cfDNA tended to be distributed as integer multiples of base pairs in a single nucleosome (~180 bp), but the saliva cfDNA did not show such a pattern (Supplementary Fig. 13a). The histone H3 pull-down experiment further demonstrated the presence of histone-DNA complexes within the serum, and possibly in higher concentration than histone-free DNA (Supplementary Fig. 13b). Together, these results demonstrated that G3@SeHANs inhibited TLR9 activation by scavenging periodontitis-derived cfDNA in vitro.

We next tested whether the related proinflammatory cytokine release in immune cells could be inhibited by G3@SeHANs. Saliva from periodontitis patients caused a prominent increase in TNF-α and IL-6 levels in RAW 264.7 macrophages (Supplementary Fig. 14), and these increases were significantly mitigated by G3@SeHANs and PAMAM-G3. We confirmed that G3@SeHANs alone without any TLR agonist showed no effect on the levels of TNF-α and IL-6, while PAMAM-G3 alone reduced the TNF-α and IL-6 expression (Supplementary Fig. 15). However, in the presence of saliva from periodontitis patients, G3@SeHANs showed a significantly greater inhibition than PAMAM-G3. We also found serum from periodontitis patients had a smaller effect on TNF-α and IL-6 levels than saliva (Supplementary Fig. 14). DAMPs from both cells and mitochondria (mtDAMPs), mtDNA, and CpG DNA caused significant increases in TNF-α and IL-6 levels in mouse and human macrophages (Supplementary Figs. 16, 17), and they were also mitigated by G3@SeHANs and PAMAM-G3 in varying degrees. Together, these results demonstrated that two cfDNA scavengers, G3@SeHANs and PAMAM-G3, were both able to inhibit periodontitis-related TLR9-mediated cellular proinflammation induced by different types of cfDNA, but also suggested that the mechanism of the inhibition might be different.

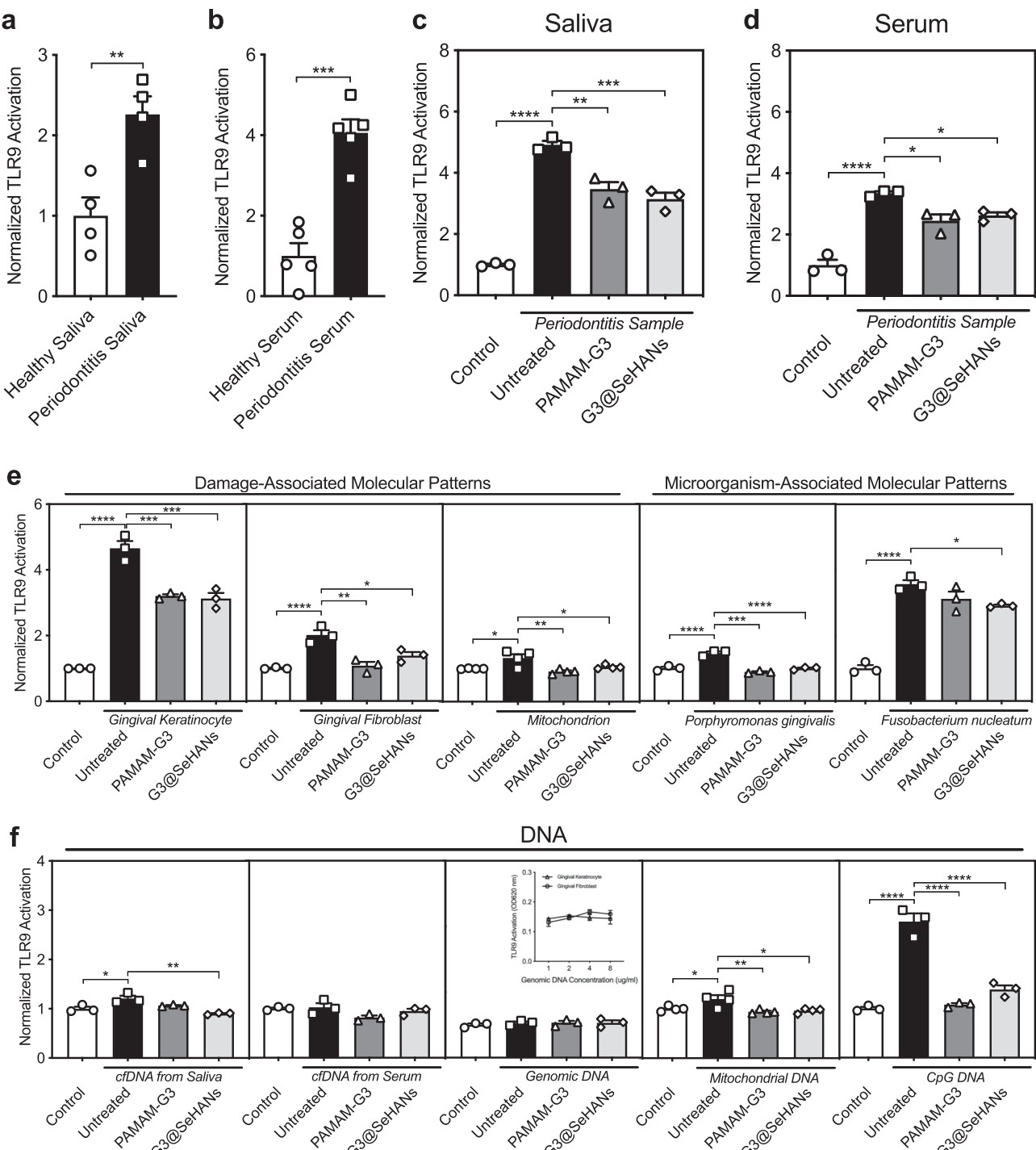

## cfDNA-scavenging mechanisms of G3@SeHANs and PAMAM-G3 are different

Due to the large differences in size, solubility, and intracellular trafficking behavior, we investigated how the cfDNA scavenging mechanisms of G3@SeHANs and PAMAM-G3 would operate differently intracellularly or extracellularly. To examine the localization of the materials, we conducted an intracellular trafficking study in RAW 264.7 cells with Cy5.5-labeled CpG DNA (Cy5.5-CpG) and FITC-labeled PAMAM-G3 or FITC-labeled G3@SeHANs. When no cationic materials were added, CpG was mainly distributed in lysosomes at 4 h and 12 h. When treated with cationic cfDNA-scavengers, both FITC-labeled PAMAM-G3 and CpG DNA were observed mainly in lysosomes and in the cytoplasm at these time points (Supplementary Fig. 18 and Fig. 4a).

In contrast, FITC-labeled G3@SeHANs were observed mainly extracellularly, and the amount of Cy5.5-CpG in lysosomes was much lower at 4 h than that of FITC-labeled PAMAM-G3. At 12 h, most Cy5.5-CpG was colocalized with FITC-labeled G3@SeHANs, and no Cy5.5 fluorescent was observed in lysosomes. This difference in CpG DNA scavenging location between G3@SeHANs (extracellular) and soluble PAMAM-G3 (intracellular) might affect their therapeutic effects.

To examine intracellular DNA-sensing signal transduction, TLR9 and downstream NF-κB pathway activation were analyzed by qRT-PCR. Saliva from periodontitis patients, mtDNA, and CpG DNA were administered as stimuli to murine macrophage RAW 264.7 cells. G3@SeHAN treatment consistently reduced the gene expression of the TLR9-NFκB pathway relative to the untreated group, whereas the

**Fig. 3 | G3@SeHANs block periodontitis-related TLR9 proinflammatory response in vitro. a**, **b** Activation of HEK-TLR9 reporter cells by healthy human saliva and periodontitis patient saliva, and activation of HEK-TLR9 reporter cells by healthy human serum and periodontitis patient serum. Data are means ± SEM; *$P < 0.05$, **$P < 0.01$, ***$P < 0.001$, ****$P < 0.0001$ by assessed by two-tailed Student's *t*-test ($n = 4$ or 5) (**a**, $P = 0.0077$; **b**, $P = 0.0002$). **c**, **d** Activation of HEK-TLR9 reporter cells by periodontitis patient saliva and periodontitis patient serum in the absence or presence of PAMAM-G3 (2 µg/mL) or G3@SeHANs (10 µg/mL) for 24 h. Data are means ± SEM; $n = 3$ independent experiments; *$P < 0.05$, **$P < 0.01$, ***$P < 0.001$, ****$P < 0.0001$ by one-way ANOVA with Tukey's multiple comparison test (**c**, $P < 0.0001$ between Control and Untreated, $P = 0.0013$ between Untreated and PAMAM-G3, $P = 0.0003$ between Untreated and G3@SeHANs; **d**, $P < 0.0001$ between Control and Untreated, $P = 0.0111$ between Untreated and PAMAM-G3, $P = 0.0333$ between Untreated and G3@SeHANs). **e** Activation of HEK-TLR9 reporter cells by DAMPs from gingival keratinocytes, gingival fibroblasts, mitochondrion, and MAMPs from *Porphyromonas gingivalis*, or *Fusobacterium nucleatum* in the absence or presence of PAMAM-G3 (2 µg/mL) or G3@SeHANs (10 µg/mL) for 24 h. Data are means ± SEM; *$P < 0.05$, **$P < 0.01$, ***$P < 0.001$, ****$P < 0.0001$ by one-way ANOVA with Tukey's multiple comparison test ($n = 4$ for mitochondrion DMAPs, $n = 3$ for the other DAMPs and MAMPs) (Gingival keratinocytes, $P < 0.0001$ between Control and Untreated, $P = 0.0004$ between Untreated and PAMAM-G3, $P = 0.0003$ between Untreated and G3@SeHANs; gingival fibroblasts, $P < 0.0001$ between Control and Untreated, $P = 0.0013$ between Untreated and PAMAM-G3, $P = 0.0150$ between Untreated and G3@SeHANs; mitochondrion, $P = 0.0153$ between Control and Untreated, $P = 0.0031$ between Untreated and PAMAM-G3, $P = 0.0416$ between Untreated and G3@SeHANs; *Porphyromonas gingivalis*, $P < 0.0001$ between Control and Untreated, $P < 0.0001$ between Untreated and PAMAM-G3, $P < 0.0001$ between Untreated and G3@SeHANs; *Fusobacterium nucleatum*, $P < 0.0001$ between Control and Untreated, $P = 0.0363$ between Untreated and G3@SeHANs). **f** Activation of HEK-TLR9 reporter cells by cfDNA from periodontitis patient saliva or serum, genomic DNA, mtDNA, and CpG DNA in the absence or presence of PAMAM-G3 (2 µg/mL) or G3@SeHANs (10 µg/mL) for 24 h. Data are means ± SEM; *$P < 0.05$, **$P < 0.01$, ***$P < 0.001$, ****$P < 0.0001$ by one-way ANOVA with Tukey's multiple comparison test ($n = 4$ for mitochondrial DNA, $n = 3$ for the other DNA) (cfDNA from saliva, $P = 0.0158$ between Control and Untreated, $P = 0.0015$ between Untreated and G3@SeHANs; mitochondrial DNA, $P = 0.0360$ between Control and Untreated, $P = 0.0073$ between Untreated and PAMAM-G3, $P = 0.0123$ between Untreated and G3@SeHANs; CpG DNA, $P < 0.0001$ between Control and Untreated, $P < 0.0001$ between Untreated and PAMAM-G3, $P < 0.0001$ between Untreated and G3@SeHANs).

effect of PAMAM-G3 was not quite stable (Supplementary Fig. 19). Together, these results suggested that the G3@SeHANs and PAMAM-G3 inhibited cfDNA-induced inflammation via different mechanisms, mostly could be attributed to scavenging cfDNA in different locations, as summarized in Fig. 4b.

## G3@SeHANs reduce inflammatory bone loss in ligature-induced periodontitis in vivo

We first determined if the cfDNA scavenging activity would be best exploited topically or systemically. We used a ligature-induced periodontitis mouse model described previously[38,39]. In this model, significant bone loss was observed approximately two weeks after the ligature, which was placed in a circle around the second molar to cause bone loss and related phenotypes, including gingival mucosa edema and tooth loosening. We tested whether targeting systemic cfDNA would be helpful to control the local inflammatory bone loss. Administered i.p. at the same dose we have used previously in a sepsis model with a different scavenger[28], we scavenged circulating cfDNA in the ligature-induced periodontitis model, but we did not observe a significant reduction of the inflammatory bone loss (Supplementary Fig. 20). Thus, we chose the topical application of nanoparticulate cfDNA scavenger for treating periodontitis (Fig. 5a). The dose of G3@SeHANs injected into the inflammatory site, 1 mg/mL, was determined in a pilot study (Supplementary Fig. 21). The four treatment groups were (1) Normal group without ligature (healthy control); (2) Untreated group with ligature and phosphate-buffered saline (PBS) injection (periodontitis control); (3) PAMAM-G3 group with ligature and PAMAM-G3 injection; and (4) G3@SeHANs group with ligature and G3@SeHANs injection. PBS, PAMAM-G3, or G3@SeHANs were injected on both sides of mice into six sites around the ligature on days 0, 3, 6, 9, and 12; samples were collected on day 15. Treatments were also smeared onto the gum around the ligature daily except on injection days to recapitulate the daily rinsing (Fig. 5b, c).

By day 15, the periodontitis-induced cfDNA and TNF-α increase in both saliva and serum was reduced by both G3@SeHANs and PAMAM-G3, with the nanoparticle more effective than the soluble counterpart (Fig. 5d–g). The inhibition effect of G3@SeHANs on saliva cfDNA and TNF-α was not only much stronger but also more sustained than that of PAMAM-G3 throughout the 15 days of treatment (Fig. 5h, i). Using micro-computed tomography (micro-CT), we observed both G3@SeHANs and PAMAM-G3 reduced ligature-induced bone loss (Fig. 5j, k). A further histopathological study by hematoxylin and eosin (H&E) staining observed that the ligatures caused a strong proinflammatory immune cell infiltration and epithelial destruction. Bone loss in the G3@SeHAN and PAMAM-G3 groups was less than in the

untreated periodontitis group, and the integrity of the epithelium, the width of alveolar bone and the degree of subepithelial immune cell infiltration was improved in the G3@SeHANs-treated group (Fig. 6a and Supplementary Fig. 22a). Osteoclastic and osteogenic activities, indicated by enzyme activity staining with tartrate-resistant acid phosphatase (TRAP) and alkaline phosphatase (ALP), respectively, were higher in the untreated periodontitis group than in normal mice. The number of TRAP+ cells was less in the G3@SeHAN and PAMAM-G3 groups than in the untreated periodontitis group (Fig. 6b and Supplementary Fig. 22b), indicating reduced osteoclastic activity. TLR9 expression was significantly higher in the untreated periodontitis group than in normal mice; both G3@SeHAN and PAMAM-G3 reduced this periodontitis-associated TLR9 expression (Fig. 6c, d). Treatment with G3@SeHANs and PAMAM-G3 also reduced TNF-α and IL-6 levels, as well as mRNA levels of the NF-κB pathway transcription factors *Rela*, *Tnf*, and *Il6* (Fig. 6e–i), indicating inhibition of NF-κB signaling.

To further determine the role of cfDNA in the pathogenesis of periodontitis and the effectiveness of the cfDNA-scavenging strategy based on G3@SeHANs, we established another periodontitis model with an additional local injection of CpG DNA (Supplementary Fig. 23a, b). CpG DNA slightly exacerbated the bone loss in comparison with the original untreated periodontitis group (Supplementary Fig. 23c, d). Both G3@SeHANs and PAMAM-G3 mitigated the increases in cfDNA and proinflammatory cytokine levels even with the influence of external CpG DNA, and G3@SeHANs showed a greater effect (Supplementary Fig. 23e). G3@SeHANs and PAMAM-G3 also reduced the severity of epithelium damage, the degree of subepithelial immune cell infiltration, and the number of osteoclasts, in comparison with CpG DNA treatment alone (Supplementary Fig. 23f, g). Together, the results demonstrated that G3@SeHANs and PAMAM-G3 alleviated cfDNA-induced inflammatory bone loss in the ligature-induced periodontitis model with G3@SeHANs showing a greater effect than PAMAM-G3. In addition, administration of G3@SeHANs did not significantly alter the diversity of the bacterial communities and microbial composition in the oral cavity of ligature-induced periodontitis mice (Supplementary Fig. 24a–g and Supplementary Table 2). Regarding safety, no gross organ damage and alteration in the systemic toxicity-related blood biochemical parameters were detected following local treatment of G3@SeHAN (Supplementary Fig. 25 and Supplementary Fig. 26).

## G3@SeHANs regulate the mononuclear phagocyte system

Macrophage polarization in gingival tissue is related to the pathogenesis of periodontitis[40,41]. M1 macrophages are proinflammatory while M2 macrophages promote repair. M1 macrophages were found

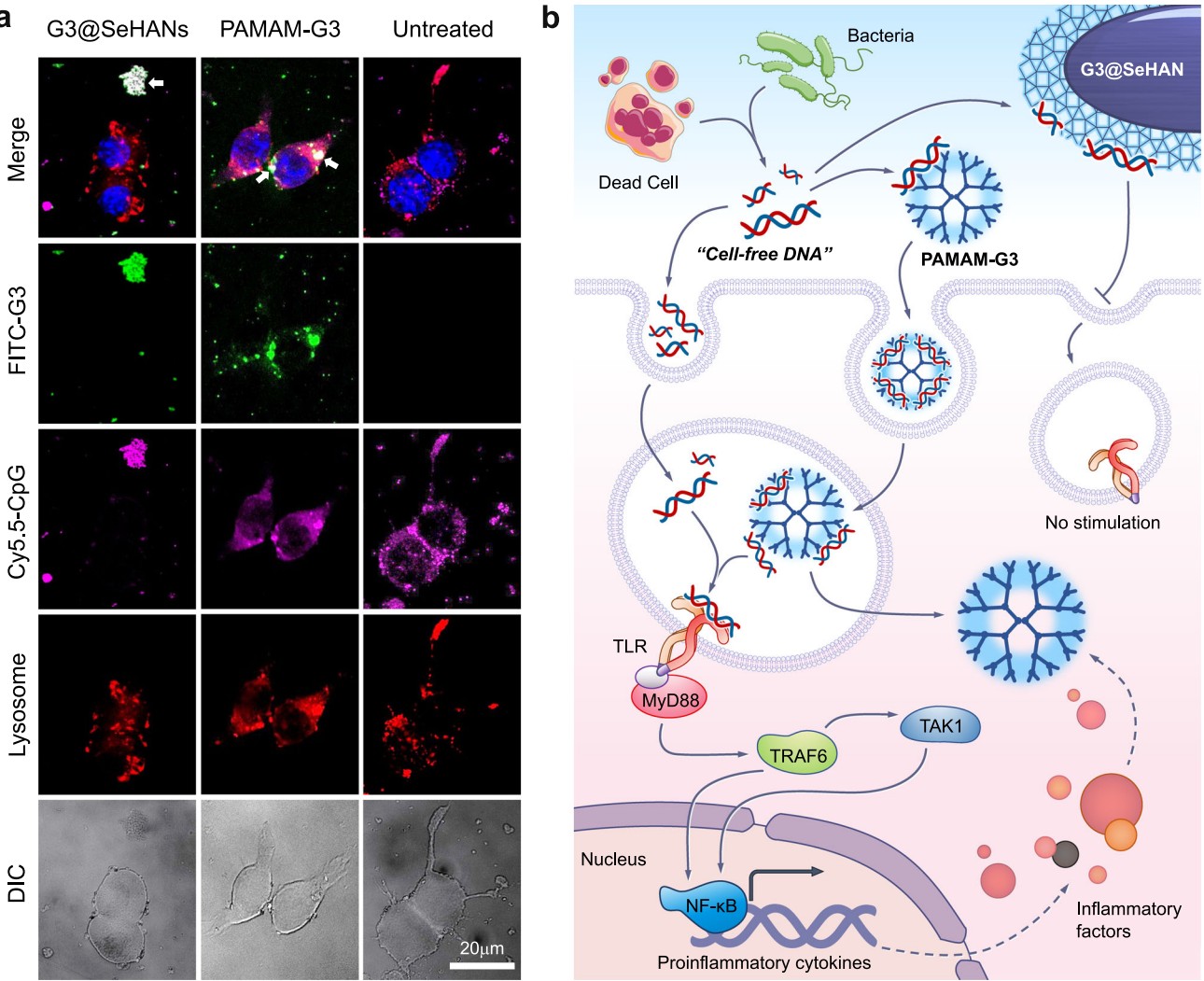

**Fig. 4 | cfDNA-scavenging mechanisms by G3@SeHANs and PAMAM-G3.**
**a** Enlarged images show intracellular localization of CpG oligonucletides and cationic materials in RAW 264.7 cells after a 12 h incubation. Scale bar, 20 µm. Colocalized CpG and cationic materials appear as white spots, and are indicated by arrows. Representative images were from 3 independent experiments. **b** cfDNA-scavenging mechanisms of G3@SeHANs and PAMAM-G3.

in the untreated periodontitis mice but not in the normal mice. Both G3@SehANs and PAMAM-G3 treatment of periodontitis mice reduced the number of M1 macrophages while increasing the number of M2 macrophages (Supplementary Fig. 27). However, the G3@SeHANs did not affect the number of dendritic cells (DC) in the periodontal tissue (Supplementary Fig. 28). T-cell-mediated host immunity is the dominant immune response in the pathogenesis of periodontitis[42]. In the in vivo study, the numbers of T cells in periodontal tissues were elevated in the untreated periodontitis group versus the normal mice group, and treatment with G3@SeHANs mitigated the increase in T cells (Supplementary Fig. 29). The adaptive T cell immune response showed a similar pattern as macrophage polarization.

To make the study more clinically relevant, we further examined the differentiation of human monocyte THP-1 cells in different conditions to investigate the effects of periodontitis saliva and G3@SeHANs treatment on the mononuclear phagocyte system. THP-1 cells were pretreated with phorbol myristate acetate (PMA) to induce differentiation from monocytes to M0-type macrophages[43]. The morphology and surface markers of cells were altered by treatment with saliva from periodontitis patients (Supplementary Figs. 30, 31). The percentage of CD197+ cells (M1 marker) was higher in response to periodontitis saliva than to healthy saliva, whereas the percentage of CD36+ cells (M2 marker) was lower (Fig. 7a–c, Supplementary Fig. 32). These

results indicated that periodontitis saliva was more prone to promote inflammation than healthy saliva. The percentages of CD83+ and CD86+ cells (DC markers) were significantly higher in response to periodontitis saliva than healthy saliva (Supplementary Fig. 33). Together, the macrophage polarization and dendrite development responded similarly to treatment with saliva from periodontitis patients.

We then tested whether G3@SeHANs and PAMAM-G3 regulated this macrophage polarization and the functions of these macrophage subsets. For the M1 phenotype, levels of TNF-α and IL-6 increased following exposure to periodontitis saliva, but this increase was significantly reduced by treatment with G3@SeHANs and PAMAM-G3 (Supplementary Fig. 34). For the M2 phenotype, exposure to periodontitis saliva increased IL-10 and reduced BMP-2 secretion, but had no effect on the TGF-β level; treatment with G3@SeHANs and PAMAM-G3 further increased the IL-10 level and also increased TGF-β and BMP-2 levels.

Periodontitis saliva significantly reduced the percentages of CD14+, CD68+, and CD36+ cells versus healthy saliva, indicating a nudge of the monocytes and M0-type macrophages toward an inflammatory status but away from the M2 phenotype. The addition of G3@SeHANs and PAMAM-G3 profoundly inhibited the reduction in CD14+, CD68+, and CD36+ cells (Fig. 7d, e). These results demonstrated that G3@SeHANs and PAMAM-G3 inhibited the proinflammatory

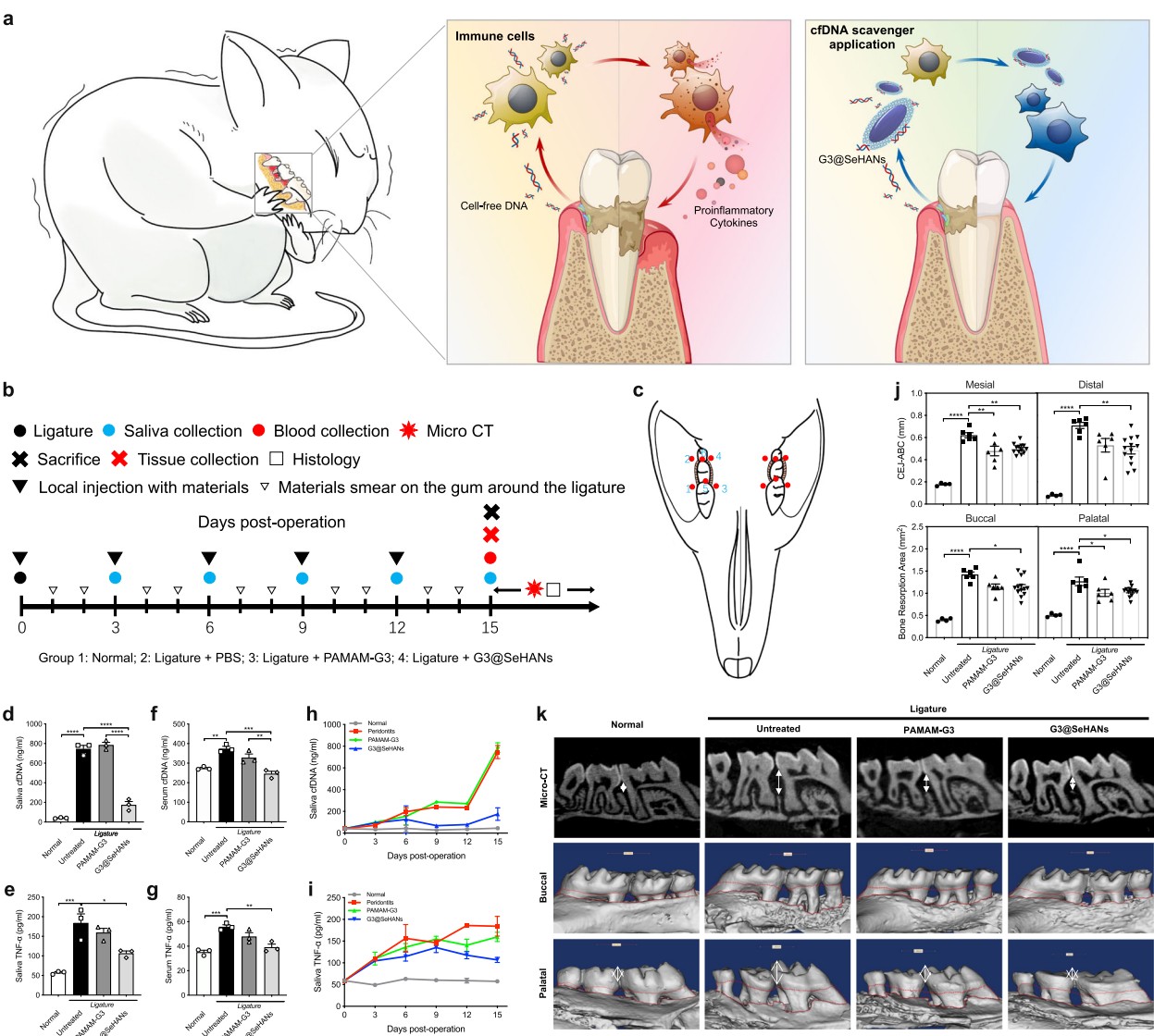

**Fig. 5 | G3@SeHANs alleviate inflammatory bone loss in ligature-induced periodontitis. a** Schematic of mechanism by which cfDNA promotes bone loss; cfDNA-scavenging nanoparticles may prevent bone loss and alleviate periodontitis. **b, c** Experimental schedule of in vivo study. PAMAM-G3 (200 µg/mL), G3@SeHANs (1 mg/mL), or PBS (control group) were administered locally by injection with a microsyringe into gingival tissue at 5 µL per site at 6 sites around the ligature: the mesiobuccal gingiva, distobuccal gingiva, mesiopalatal gingiva, distopalatal gingiva, and mesial and distal gingival papilla. Microinjections were performed on days 0, 3, 6, 9, and 12. On experimental days without injections, materials or PBS were noninvasively smeared on the gingiva with disposable microapplicators. Both sides of the mice were treated with the same materials. Levels of (**d**) saliva cfDNA, (**e**) saliva TNF-α, (**f**) serum cfDNA, and (**g**) serum TNF-α at 15 days post-operation. Data are means ± SEM; *n* = 3 samples per group; *P < 0.05, **P < 0.01, ***P < 0.001, ****P < 0.0001 by one-way ANOVA with Tukey's multiple comparison test (**d**, P < 0.0001 between Normal and Untreated, P < 0.0001 between Untreated and G3@SeHANs, P < 0.0001 between PAMAM-G3 and G3@SeHANs; **e** P = 0.0006 between Normal and Untreated, P = 0.0124 between Untreated and G3@SeHANs;

**f** P = 0.0023 between Normal and Untreated, P = 0.0004 between Untreated and G3@SeHANs, P = 0.0078 between PAMAM-G3 and G3@SeHANs; **g** P = 0.0009 between Normal and Untreated, P = 0.0035 between Untreated and G3@SeHANs). **h, i** Changes in cfDNA and TNF-α concentrations in the saliva of mice. Data are means ± SEM; *n* = 3 samples per group. **j** Bone loss measured by the vertical distance between the cementoenamel junction (CEJ) and alveolar bone crest (ABC) (CEJ-ABC) and the bone resorption area. Data are means ± SEM; *n* = 4 for normal group, *n* = 6 for untreated and PAMAM-G3 groups, *n* = 13 for G3@SeHANs group; *P < 0.05, **P < 0.01, ****P < 0.0001 by one-way ANOVA with Tukey's multiple comparison test (Mesial, P < 0.0001 between Normal and Untreated, P = 0.0028 between Untreated and PAMAM-G3, P = 0.0026 between Untreated and G3@SeHANs; Distal, P < 0.0001 between Normal and Untreated, P = 0.0027 between Untreated and G3@SeHANs; Buccal, P < 0.0001 between Normal and Untreated, P = 0.0151 between Untreated and G3@SeHANs; Palatal, P < 0.0001 between Normal and Untreated, P = 0.0399 between Untreated and PAMAM-G3, P = 0.0339 between Untreated and G3@SeHANs). (**k**) Micro-CT scanning and 3D reconstruction of the bone loss.

polarization of macrophages. Thus, G3@SeHANs can regulate the mononuclear phagocyte system to an anti-inflammatory and repair-favorable status for periodontitis treatment.

## Discussion

Dysregulation of proinflammatory activation triggers imbalances in the innate immune system, leading to periodontitis[1]. TLR9-mediated

proinflammation is one of the novel proinflammatory pathways for DNA sensing in periodontal disease pathogenesis[44]. cfDNA, serving as the ligand to TLR9, is the collection of endogenous DNA released by damaged host cells, and exogenous bacterial or viral DNA[13,14]. cfDNA in body fluids can activate TLR9 signaling through unmethylated CpG motifs (predominantly found in mitochondrial DNA and bacterial DNA) or through phosphodiester backbone in a sequence-

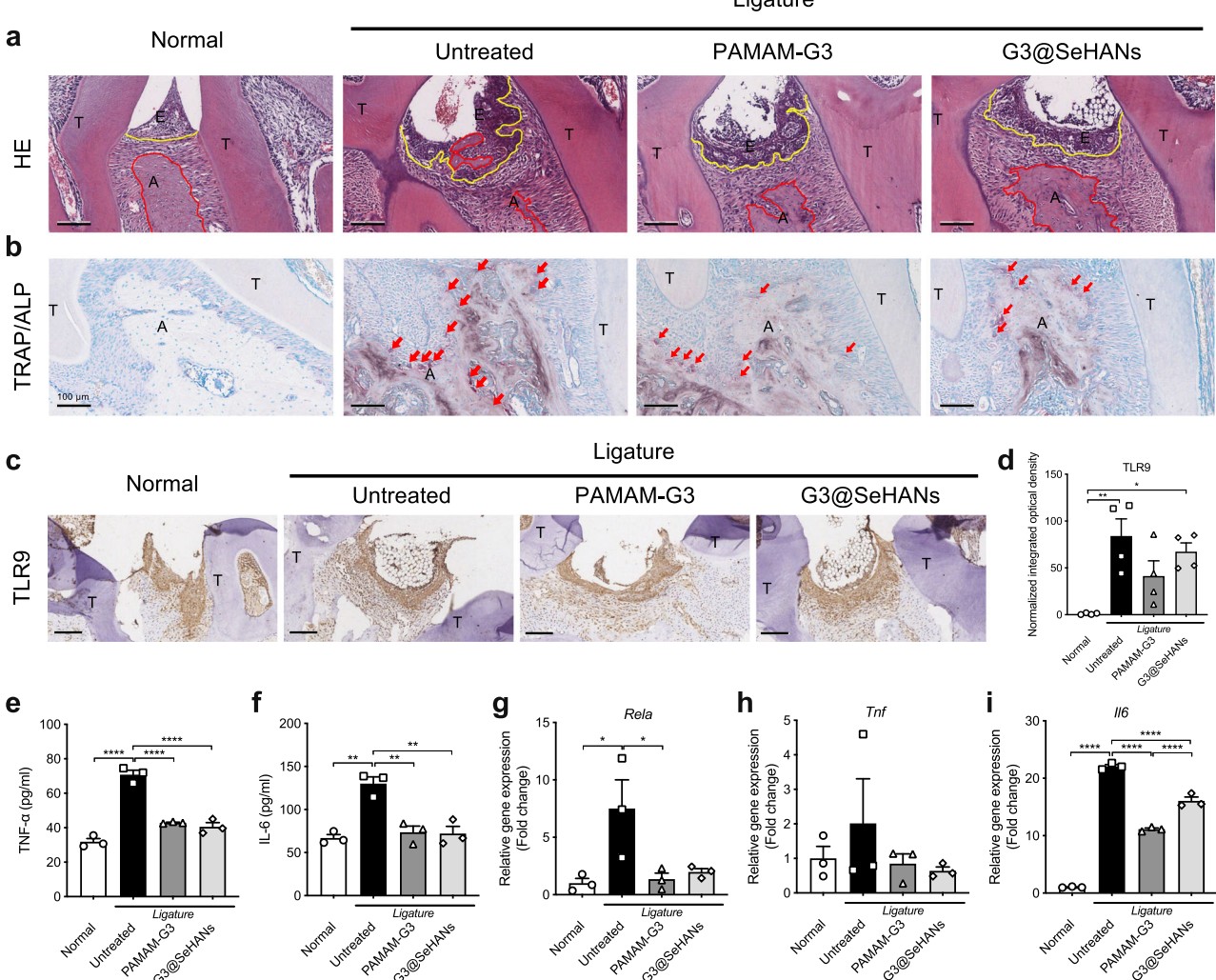

**Fig. 6 | G3@SeHANs inhibit cellular proinflammatory response in ligature-induced periodontitis. a** H&E staining of periodontal tissues on day 15 after G3@SeHAN administration. Scale bars, 100 μm. Inflammatory cell infiltration into the epithelium and bone destruction were evident in the untreated model, whereas G3@SeHAN treatment prevented these pathological changes. (E, epithelium; A, alveolar bone; T, tooth; yellow line indicates epithelial basement membrane; red line indicates alveolar bone crest). **b** TRAP/ALP staining of periodontal tissues on day 15 after G3@SeHAN administration. Scale bars, 100 μm. The number of TRAP⁺ osteoclasts (red arrows) in the untreated group was higher than in the DNA scavenger-treated groups. **c** IHC staining of TLR9 in periodontal tissues on day 15 after G3@SeHAN administration. Scale bars, 100 μm. TLR9 expression increased significantly in the untreated group; treatment with G3@SeHAN or PAMAM-G3 reduced TLR9 expression. Images (**a–c**) were representative of three independent mice. **d** Expression of TLR9 in the epithelium of periodontal tissues. Data are means ± SEM; *n* = 4 samples per group; \**P* < 0.05, \*\**P* < 0.01, \*\*\**P* < 0.001, \*\*\*\**P* < 0.0001 by one-way ANOVA with Tukey's multiple comparison test (*P* = 0.0035 between Normal and Untreated, *P* = 0.0171 between Normal and G3@SeHANs). (**e–i**) TNF-α and IL-6 levels in periodontal tissues, and relative gene expression of *Rela*, *Tnf*, and *Il6* in periodontal tissues. Data are means ± SEM; *n* = 3 samples per group; \**P* < 0.05, \*\**P* < 0.01, \*\*\**P* < 0.001, \*\*\*\**P* < 0.0001 by one-way ANOVA with Tukey's multiple comparison test (**e**, *P* < 0.0001 between Normal and Untreated, *P* < 0.0001 between Untreated and PAMAM-G3, *P* < 0.0001 between Untreated and G3@SeHANs; **f** *P* = 0.0010 between Normal and Untreated, *P* = 0.0022 between Untreated and PAMAM-G3, *P* = 0.0018 between Untreated and G3@SeHANs; **g** *P* = 0.0315 between Normal and Untreated, *P* = 0.0411 between Untreated and PAMAM-G3; **i** *P* < 0.0001 between Normal and Untreated, *P* < 0.0001 between Untreated and PAMAM-G3, *P* < 0.0001 between Untreated and G3@SeHANs, *P* < 0.0001 between PAMAM-G3 and G3@SeHANs).

independent manner[45]. Some proportion of cfDNA may circulate as histone-DNA complex or other DNA-protein complex, rather than protein-free DNA[46], and evidence suggested that this type of cfDNA also could activate TLR9[47–49]. cfDNA of different origins are critical in many inflammatory diseases and elevated circulating cfDNA is found in the serum of patients with systemic diseases including rheumatoid arthritis, sepsis, atherosclerosis, and cancer[50–53]. However, whether and which kind of cfDNA contributes to periodontitis, whether it is through TLR9 mediated proinflammatory pathway, remains unclear. Given this, we systematically investigated the correlation among cfDNA, TLR9 and periodontitis, thereby fabricating a periodontitis-specific nanoparticle cfDNA scavenger, G3@SeHANs, which effectively alleviated cfDNA-mediated inflammatory alveolar bone loss.

We firstly demonstrated that cfDNA correlated with the progression of periodontitis. We found that levels of both local and circulating cfDNA were elevated in mice and patients with periodontitis. Changes in cfDNA levels in saliva during periodontitis establishment in mice were similar to those previously observed in synovial fluid during the establishment of rheumatoid arthritis in mice[54]. In patient samples, correlations of cfDNA in saliva and serum with periodontitis were demonstrated.

As the strategy of scavenging cfDNA by nucleic acid-binding nanoparticles shows the therapeutic efficacy in other circulating or localized inflammatory diseases like sepsis and rheumatoid arthritis[28,54,55], the cfDNA-scavenging strategy appears to be a potential therapeutic approach to treat periodontitis. Herein, we

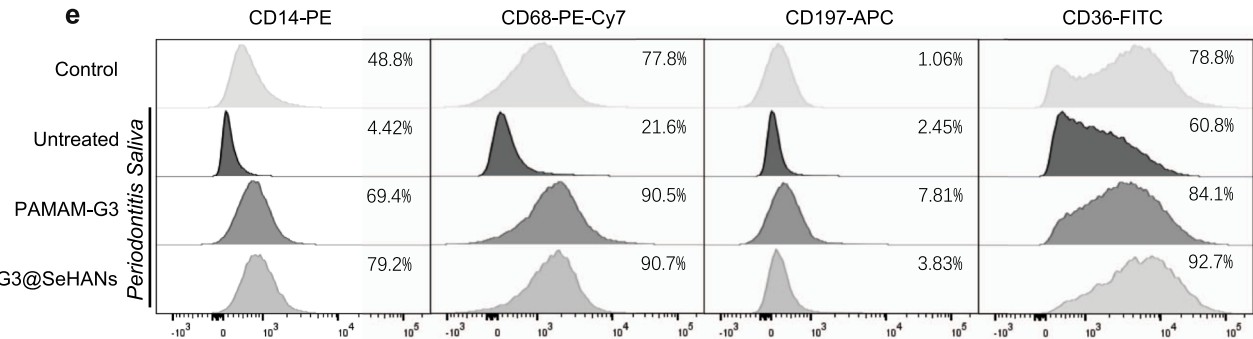

**Fig. 7 | Effects of G3@SeHANs on macrophage polarization. a–c** Expression of the M1 marker CD197 and the M2 marker CD36 in THP-1 cells activated by periodontitis saliva. Data are means ± SEM; differences were assessed by two-tailed Student's *t*-test (*n* = 3 samples per group) (**c**, *P* = 0.0011). **d** Schematic of macrophage polarization by periodontitis saliva and role of cationic cfDNA scavengers. **e** Effects of periodontitis saliva and cationic scavengers on expression of CD14, CD68, CD197, and CD36 in THP-1 cells. *\*P* < 0.05, *\*\*P* < 0.01.

fabricated the periodontitis-specific nanoparticulate cfDNA scavenger, G3@SeHANs. As a functional component of G3@SeHANs, PAMAM-G3 is a widely studied soluble nucleic acid binding polymer that can neutralize proinflammatory TLR9 agonists, but its toxicity hinders its clinical translation. Such toxicity of soluble PAMAM-G3 was attenuated by coating on SeHANs to generate insoluble G3@SeHANs. More importantly, according to the TG analysis, the mass percentage of PAMAM-G3 in G3@SeHANs was only 2%, which means there were only 0.2 μg of PAMAM-G3 in 10 μg of G3@SeHANs. Nonetheless, 10 μg/mL of G3@SeHANs matched the cfDNA-scavenging activity of 2 μg/mL of the much more toxic free PAMAM-G3. These results demonstrate the translational potential of the G3@SeHANs.

We chose SeHANs as the core of G3@SeHANs mainly for the following reasons. Element-substituted hydroxyapatite is ubiquitous in the natural hard tissue of vertebrates, making it favorable for clinical translation in orthopedic and dental diseases. Selenium is an essential nutrient that helps the maintenance of homeostasis[56]. Blood selenium is inversely correlated with the rate of bone turnover[57]. Mechanistic studies have suggested that the biological functions of selenium mainly involve an antioxidant defense[33,34]. In this study, the SeHANs were highly ordered with a uniaxially oriented hierarchical structure similar to that of Bio-Oss®, a collagen-free natural bone apatite that is commonly used in alveolar bone repair (Fig. 2a and Figs. S5, S8)[58,59]. Our previous work demonstrated that the synthetic SeHANs not only showed good biocompatibility and promoting effects in osteogenic

differentiation of mesenchymal stem cells, but also induced autophagy and apoptosis in osteosarcoma cells for bone tumor inhibition[35,36]. Therefore, the nanoparticulate cfDNA-scavenger was built on a SeHANs core to maximize the chance of alleviating inflammatory alveolar bone loss.

We first showed that in vitro G3@SeHANs could reduce DNA-mediated TLR9 activation triggered by synthetic DAMPs well as by a collection of complex PAMPs and DAMPs in the periodontitis saliva and periodontitis serum. Following it up with in vivo validation, we initially applied the mesoporous silica nanoparticles functionalized with polyethyleneimine (MSN-PEI), a nanoparticulate scavenger proven fruitful in tackling the systemic infection in sepsis[28] via intraperitoneal injection. While this could reduce the cfDNA level in the systemic circulation, it could not alleviate the local bone loss significantly after 15 days. Therefore, we decided to focus on topical application. Regarding topical application for periodontitis, there are many examples in the clinic, for example, treatment of the gingival sulcus with minocycline hydrochloride ointment to control the local inflammation after dental scaling[60]. However, the space of the gingival sulcus in mice is too small for nanoparticle injection. Thus, a topical application was applied, which included injecting the materials into the periodontal tissue and smearing on the gingiva.

We carried out two in vivo studies to validate the hypothesis. First, we used G3@SeHANs to treat ligature-induced periodontitis to test the cfDNA-scavenging strategy. Local injection of G3@SeHANs into the periodontal tissue not only decreased the cfDNA level in saliva but also in serum. TLR9 expression in the gingival epithelium was significantly greater than that in normal epithelium and was also reduced by G3@SeHANs. Second, we injected CpG DNA into the periodontal tissue to elucidate the function of the G3@SeHANs. A slight increase in alveolar bone loss in the CpG DNA-enhanced untreated group was found. We noticed that the distance of the cemento-enamel junction to the alveolar bone crest (CEJ-ABC) and bone resorption areas of the untreated group in the second in vivo study (with CpG DNA injection) were not the same as in the first study, possibly due to a difference between different batches of mice. Nevertheless, the G3@SeHANs consistently inhibited periodontitis and reduced levels of local and circulating cfDNA and proinflammatory factors. Taken together, the results suggest that scavenging cfDNA to block hyperactive proinflammatory response is useful for treating uncontrolled localized inflammatory bone loss in periodontitis.

The correlation between periodontitis and systemic diseases has been an active area of research for several decades[61]. Studies have demonstrated that cfDNA appears to link periodontitis with systemic diseases, as periodontal bacterial DNA has been found in serum, cardiac tissue, synovial fluid, and the intrauterine environment in these diseases[10,62,63]. Circulating cfDNA plays an important role in many systemic diseases, including rheumatoid arthritis, sepsis, atherosclerosis, and cancer[50–53,64]. Therefore, understanding whether the accumulation of circulating cfDNA corresponds to local inflammation is important.

One interesting finding in this study was that a localized cfDNA blockade influenced the circulating cfDNA level. Evidence in this study could support that in periodontitis, the change of circulating cfDNA in serum was the consequence of the local inflammation. It could be a causality that the decrease of local cfDNA level led to a less severe local inflammation. As discussed above, the cfDNA levels in both saliva and serum were elevated in periodontitis, but the cfDNA level in saliva showed a stronger correlation with periodontitis than the cfDNA level in serum. PLI > 2, BI > 2, and PD > 4 mm are remarkable and reliable parameters for measuring periodontal inflammation, and the correlations of cfDNA in saliva were the strongest in terms of these respective parameters, but cfDNA in serum did not show these patterns. It is tempting to speculate that the inflammation originated locally and spread to the systemic level as inflammation progressed, which would explain why scavenging the localized cfDNA would alleviate the bone loss and influence the circulating cfDNA level.

We attempted to understand what was the major player that caused proinflammation. cfDNA from saliva, mtDNA, and CpG DNA all activated the TLR9 pathway. The saliva and serum from periodontitis patients enhanced TLR9 activation, but that activation by saliva was much stronger than by serum. During the innate immune response, MAMPs continuously accumulate because of dying bacteria[65]. Local inflammation also leads to cell death and increases the level of DAMPs[66]. Thus, in the periodontal inflammatory microenvironment, cfDNA could be a collection of both MAMPs-derived DNA and DAMPs-derived DNA. In rheumatoid arthritis, CpG motifs in DNA have been proven to correlate with TLR9 activation and inflammatory bone loss[51]. In periodontitis, the saliva is constantly exposed to microbes, therefore a source of abundant CpG-motif-rich bDNA. Thus CpG-motif-rich bDNA and mtDNA may be the reason why saliva cfDNA induces TLR9 activation strongly[67,68]. For circulating cfDNA, a significant proportion is bound to protein molecules such as nucleosomes, a DNA-histone complex[69]. TLR9 activation by DNA sometimes needs the help of protein[47–49], and protein-induced curvature of DNA ligands can promote ligand recognition by TLR9[70]. To confirm the existence of nucleosome in cfDNA collection, we examined the cfDNA fragments from serum and saliva, which showed distinct size characteristics. The serum-extracted cfDNA possessed a specific distribution pattern of DNA fragment size (~180 bp), which manifested as integer multiples of base pairs in a single nucleosome. However, there was no such distinct distribution pattern for cfDNA from saliva. Further, highly fragmented DNA could be detected on magnetic beads after the histone H3 pull-down of the serum sample. This fraction of DNA, as histone-DNA complex, showed a higher concentration compared to uncaptured histone-free DNA, and retained some of the size distribution characteristics same as nucleosome DNA. These patterns suggested that cfDNA within the serum might be derived from endogenous genomic DNA released by dead host cells, and was presented as extracellular nucleosomes, but the saliva could have a larger proportion of exogenous origin (like bacterial) of cfDNA rather than nucleosome. For serum cfDNA, the extraction process of DNA would remove proteins, which could break the structure of the nucleosome. Together, it explained why cfDNA from saliva showed stronger activating capacity on TLR9 reporter cells than cfDNA from serum.

Although both G3@SeHANs and PAMAM-G3 reduced proinflammation, differences between the effects of these two cfDNA-scavenging materials were evident. PAMAM-G3 is soluble and can be internalized by cells via pinocytosis, whereas the large G3@SeHANs remain extracellular. A strength of our study is that we confirmed differences in therapeutic activity between a soluble, internalized nucleic acid-binding polymer and an insoluble, extracellular nucleic acid-binding nanoparticle. Based on qRT-PCR analysis of DNA-sensing signaling, the larger, insoluble G3@SeHANs appeared to achieve a more stable depletion of danger signals. Another important aspect that requires further study is what happened to the cfDNA absorbed by the PAMAM-G3. Soluble PAMAM-G3 scavenges cfDNA intracellularly and can also be used to deliver cfDNA into cells in gene delivery. Intracellularly scavenged cfDNA could further activate the proinflammatory pathway. Extracellular nucleic acid-binding nanoparticles might avoid this possibility because the cfDNA is scavenged before cell entry, preventing interactions between the agonists and the TLR receptors.

In previous studies, we showed that selenium was released from SeHANs as the doped hydroxyapatite biodegraded, and that selenium had an antibacterial effect[31,36]. Here we wanted to avoid disrupting the balance of microbiota in a study related to oral or gut research because serious complications may occur when this balance is disrupted. 16 S RNA analysis showed that G3@SeHANs did not influence the oral microbiota community, a favorable result in view of the theory that a

smaller influence on the microbiota is indicative of a better medical intervention[71]. The lack of disruption of microbiota could be due to the injection of G3@SeHANs into tissues; any selenium released from the tissue may not influence the balance of oral surface microbiota. We did not include the normal mice group (without ligature) in the 16 S RNA analysis because the ligature around the tooth would change the microbiota in the oral cavity of mice.

Macrophages in the mononuclear phagocyte system play an important role in periodontitis[40,72,73]. Clinical observations have shown that M1/M2 macrophages are correlated with the stage of periodontitis. Our in vivo results are consistent with these clinical findings. The previous study of cfDNA-scavenging strategy to treat sepsis showed that cfDNA promoted the proinflammatory phenotype of macrophages[28]; we obtained similar results here. We showed that periodontitis saliva contained more inflammatory stimulatory components than normal saliva. Like the in vivo results of the sepsis study[28], here, G3@SeHANs and PAMAM-G3 did reduce the M1 phenotype and promote the M2 phenotype. Our results again suggest that macrophage polarization can be modulated by the cfDNA-scavenging strategy.

The superiority of the two animal models, namely ligature-induced or periodontal bacterial-induced periodontitis (LIP and BIP), remains unresolved. Comparatively, the BIP model shares more similar pathogenic factors and course of disease with human chronic periodontitis. However, a long period is needed for the onset of BIP, and the degree of bone loss is unstable. Differences between batches and individuals exist continuously. In contrast, the LIP model has a shorter onset period and a more stable and severe bone loss phenotype. In rodent models, the influence of trauma on the pathogenesis of LIP cannot be excluded. However, a study specifically devoted to the establishment of LIP models showed that the administration of antibiotic treatment to LIP mice significantly alleviated the alveolar bone loss[39]. It suggests that the dysbiotic oral microbiota still plays an important role in the pathogenesis of LIP. In addition, since tartar (mineralized plaque, soft scale, and food residue around the gingival sulcus) is one of the main clinical manifestations and pathogenic factors of periodontitis, the traumatic damage of the local gingival barrier could also be one of the driving factors for the progression of periodontitis. LIP as an animal model has also been chosen for the study on the local pathological immune response in periodontitis, and the results show that the periodontitis patients shared similar oral mucosal immunopathology with the LIP model[74]. Thus, based on the stability of the model, the significance of the phenotype and at least the partial consistency between the LIP model and the periodontitis, LIP was selected as the animal model used in our study. Besides, the scavenging of DAMPs was also evaluated in vitro with comprehensive consideration of the traumatic factor.

When we first conducted the pilot study on the LIP model, we injected the nanoparticulate cfDNA scavenger into the gingival tissue every other day. Unfortunately, the tissue would be swollen. In addition, the frequency appeared too high as the anesthesia required for injection seemed to weaken the animals. We, therefore, decreased the injection frequency to once every three days and smeared the nanomaterials on the gingiva every day, which would simulate the mouth-rinsing action and it produced a positive outcome. Should this technology be translated, topical application of nanomaterials to the gingival sulcus would not be a problem since there is ample space in the human gingival sulcus. In the clinic, the practice of topical application to the gingival sulcus for controlling periodontal inflammation, such as subgingival minocycline hydrochloride ointment, has been around for many years[60]. Our frequency of once every three days is lower than the application of chamomile and lidocaine hydrochloride gel in the clinic. It suggests a potential for translation because the formulation of ointment with these cfDNA-scavengers should not be difficult[75].

In summary, this study revealed the cfDNA level correlated with the progression of periodontitis and could be a new target for periodontitis treatment. The cfDNA-scavenging approach shows promising potential for alleviating periodontitis. To harness the proven therapeutic activity of soluble PAMAM-G3 polymer while reducing its toxicity, we developed a periodontitis-specific nanoparticulate cfDNA scavenger consisting of bone-mimicking selenium-doped hydroxyapatite coated with PAMAM-G3 that favorably alleviated proinflammation, resulting in inhibition of cfDNA-mediated inflammatory alveolar bone loss. Together, our findings suggest a viable cfDNA-scavenging approach that is safe, effective, and clinically translatable for treating periodontitis.

## Methods
### Materials and reagents
Calcium nitrate tetrahydrate, trisodium phosphate, sodium selenite, linoleic acid, octadecylamine, anhydrous ethanol, nitrate acid, PAMAM-G3, anti-TLR9 rabbit polyclonal antibody (cat. #SAB3500313), fluorescein isothiocyanate isomer I (FITC), 4′,6-diamidine-2′-phenylindole dihydrochloride (DAPI), brain-heart infusion broth (BHI), agar, hemin and menadione, cell counting kit-8 (CCK-8), and PMA were purchased from Sigma-Aldrich (St. Louis, MO, U.S.A.). Poly(I:C) (tlrl-picw), ultrapure LPS (tlrl-peklps), ORN06/LyoVec (tlrl-orn6), ODN BW006 (tlrl-bw006), ODN 2395, and QUANTI-Blue medium were purchased from InvivoGen (San Diego, CA, U.S.A). A bicinchoninic acid (BCA) protein assay kit, H&E staining kit, defibrinated sheep blood, and Schaedler broth were purchased from Solarbio (Beijing, China). Paraformaldehyde (4%) was purchased from Servicebio (Wuhan, Hubei, China). A TRAP/ALP staining kit was purchased from FUJIFILM Wako Chemicals (Osaka, Japan). Anti-iNOS (D6B6S) rabbit monoclonal antibody (cat. #13120), anti-arginase-1 (D4E3M) XP rabbit monoclonal antibody (cat. #93668), and PathScan Sandwich ELISA Lysis Buffer were purchased from Cell Signaling Technology (Danvers, Massachusetts, U.S.A.). Alexa Fluor 594-conjugated goat anti-rabbit IgG (H + L) (cat. #ZF-0516) was purchased from ZSGB-BIO (Beijing, China). Anti-CD11c (N418) Armenian hamster monoclonal antibody (cat. #14-0114-82), anti-F4/80 (BM8) rat monoclonal antibody (cat. #14-4801-82), APC anti-human CD83 monoclonal antibody (HB15e) (cat. #17-0839-4), PE anti-human CD86 monoclonal antibody (IT2.2) (cat. #12-0869-41), PE-Cy7 anti-human CD209 monoclonal antibody (eB-h209) (cat. #25-2099-41), PE anti-human CD14 monoclonal antibody (61D3) (cat. #12-0149-41), PE-Cy7 anti-human CD68 monoclonal antibody (Y1/82 A) (cat. #25-0689-41), and APC anti-human CD197 monoclonal antibody (3D12) (cat. #17-1979-41) were purchased from eBioscience (San Diego, CA, U.S.A.). An anti-rabbit HRP-DAB IHC Detection Kit (cat. #CTS005) was purchased from Novus (Minneapolis, MN, U.S.A.). FITC anti-human CD14 monoclonal antibody (HCD14) (cat. #325603) was purchased from BioLegend (San Diego, California, U.S.A.). FITC anti-human CD36 monoclonal antibody (CB38) (cat. #555454) was purchased from BD Biosciences. Alexa Fluor 647-conjugated goat anti-Armenian hamster IgG antibody (cat. #ab173004), anti-CD3 rabbit polyclonal antibody (cat. #ab5690), and ChIP grade anti-histone H3 antibody (cat. #ab1791) were purchased from Abcam (Cambridge, UK). Dylight 488-conjugated AffiniPure goat anti-rat IgG (H + L) (cat. #E032240) was purchased from EARTHOX (San Francisco, CA, U.S.A.). ProLong Gold Antifade Mountant (P36930), UltraPure salmon sperm DNA, a Quant-iT PicoGreen dsDNA assay kit, LysoTracker Red DND-99, a Mitochondria Isolation Kit, ELISA kits for human and mouse TNF-α, human and mouse IL-6, human IL-10, human TGF-β, and BMP-2, a TaqMan Advanced miRNA cDNA Synthesis Kit, and a RevertAid First Strand cDNA Synthesis Kit were purchased from Thermo Scientific (Waltham, Massachusetts, U.S.A). An iScript One-Step RT-PCR Kit with SYBR Green was purchased from Bio-Rad (Hercules, CA, U.S.A.). Dulbecco's modified Eagle's medium (DMEM), RPMI-1640 medium, fetal bovine serum (FBS), and 0.25% trypsin-EDTA were purchased from Gibco

(Carlsbad, CA, U.S.A.). A dermal cell basal medium and keratinocyte growth kit were purchased from the American Type Culture Collection (ATCC, Manassas, VA, U.S.A.). An RNApure Total RNA Fast Extraction Kit (cat. #RP1202) was purchased from Bioteke Corporation (Wuxi, Jiangsu, China). Cy5.5-labeled CpG 1826 and primers (Supplementary Table 3) were synthesized and purchased from Integrated DNA Technologies (IDT, Coralville, IA, U.S.A.). All other reagents were commercially available and were used as received.

## Synthesis of SeHANs
SeHANs were synthesized using a modified liquid-solid-solution (LSS) method. Briefly, 1.18 g calcium nitrate tetrahydrate was dissolved in 25 mL deionized water and then mixed with an organic solution composed of 1.5 g octadecylamine, 12 mL linoleic acid, and 48 mL anhydrous ethanol. Thereafter, 461.25 mg trisodium phosphate and 48.65 mg sodium selenite were dissolved in 20 mL deionized water and added dropwise to the above mixture. After stirring at room temperature for 10 min, the suspension was transferred into a hydrothermalactor and allowed to react for 12 h at 110 °C. The resulting precipitates were washed with anhydrous ethanol and deionized water at least six times and collected by centrifugation, and were stored in anhydrous ethanol at 4 °C.

## Synthesis of G3@SeHANs
To decorate SeHANs with PAMAM-G3 (G3), SeHANs (1 mg) were mixed with G3 (10 mg) in PBS (pH 7.4, 0.2 mL), and the mixture was incubated for 12 h at room temperature with shaking. G3-coated SeHANs (G3@SeHANs) were centrifuged at 845 g for 5 min to remove unbound G3 and were washed three times with PBS. The size and zeta potential of G3@SeHANs were measured with a Zetasizer (Nano ZS90, Malvern Panalytical).

## Characterization of SeHANs and G3@SeHANs
XRD (PANalytical B.V., Holland) and FTIR (Vertex 70, Bruker, Germany) analyses were performed to investigate the phase composition and functional structure of natural bone apatite (Bio-Oss®), SeHANs and G3@SeHANs. Morphology studies were performed using field emission scanning electron microscopy (FSEM, FEI, Holland) and field emission transmission electron microscopy (FTEM, FEI, Holland). Elemental mapping was performed using FTEM. The valence state of selenium in nanoparticles and PAMAM-G3 grafting on the surface of SeHANs were determined by XPS (Kratos, Japan). PAMAM-G3 content was detected by thermogravimetric analysis (TGA, PerkinElmer, U.S.A.) in an $N_2$ atmosphere. Specific surface area was measured by BET analysis (Micromeritics, U.S.A.).

## Selenium content in SeHANs and G3@SeHANs
The amount of selenium in synthesized nanoparticles was quantified using an inductively coupled plasma optical emission spectrometer (ICP-OES, Prodigy Plus, Leeman Labs, U.S.A.). Nanoparticles (2.5 mg) were dissolved in 1 mL of 70 wt% nitric acid and diluted to 10 mL with 1 wt% nitric acid before ICP-OES analysis.

## Patient sample collection
Saliva and serum samples for clinical correlation analysis were obtained from 10 healthy volunteers, 10 gingivitis and 10 periodontitis patients from West China Hospital of Stomatology, Sichuan University. Patient sample collection was performed with the approval of the Ethics Committee of West China Hospital of Stomatology, Sichuan University (WCHSIRB-D-2020-461). All participants in this study signed an informed consent form prior to sample collection and were informed of the potential benefits and risks of participating. Gingivitis and periodontitis were diagnosed according to the 2018 new classification and definition of periodontitis[76], as well as the definition of periodontal clinical healthy control. Demographic information and a complete periodontal clinical examination of the whole dentition were recorded for all participants by the same periodontist. Periodontal clinical parameters recorded during the examination included PLI[77], BI[78] and PD. As participant compensation, participants who completed sample collection received a complete oral examination and, if needed, periodontal treatment for free.

## Biodegradation and selenium release by SeHANs and G3@SeHANs
Selenium release by SeHANs and G3@SeHANs, which represented their biodegradation, was detected by ICP-OES in three different solutions: PBS pH 7.4, saliva from healthy donors, and saliva from periodontitis patients. Saliva was collected and sterilized with a 0.22 μm filter (Millipore), then nanoparticles were prepared as a suspension at 2.5 mg/mL with these three solutions in 1.5 mL tubes. Thereafter, the tubes were placed in a water bath shaker at 37 °C. At each time point, the tubes were centrifuged at 14000 g for 15 min to collect 1 mL of supernatant, and 1 mL of fresh buffer solution was added to restore the volume. The collected supernatant was digested with 1 mL of 70 wt% nitric acid and diluted to 10 mL with 1 wt% nitric acid before ICP-OES analysis.

## DNA-binding efficiency of SeHANs and G3@SeHANs
Bare SeHANs or G3@SeHANs (200 μg) were mixed with salmon sperm DNA (2–30 μg in 1 mL Tris-EDTA buffer) and incubated for 3 h at room temperature with shaking. The mixture was centrifuged at 845 g for 5 min, and the supernatant containing unbound DNA was collected and analyzed with a Quant-iT PicoGreen dsDNA assay to determine the amount of DNA bound to the nanoparticles.

## Bacterial culture
*Porphyromonas gingivalis* and *Fusobacterium nucleatum* strains were provided by the State Key Laboratory of Oral Diseases, the West China School of Stomatology, Sichuan University. *Porphyromonas gingivalis* and were stored in glycerol broth at −80 °C. The bacteria were cultured in brain heart infusion (BHI) or 5% defibrinated sheep blood agar, both supplemented with 5 mg/mL hemin-menadione. *Fusobacterium nucleatum* was cultured in Schaedler broth. Bacteria were incubated in an anaerobic environment (90% N, 5% CO, 5% $H_2$) at 37 °C.

## DAMP and MAMP isolation
HGF-1 (cat. #CRL-2014) and primary gingival keratinocytes (PGKs) (cat. #PCS-200-014) were purchased from American Type Culture Collection (ATCC) (Manassas, VA) and were maintained according to ATCC protocols and were harvested for DAMP collection. Mitochondria were collected from HGFs with a mitochondrion isolation kit and were used for DAMP isolation. Bacteria (*Porphyromonas gingivalis* and *Fusobacterium nucleatum*) were used for MAMP isolation. Cells and bacteria were collected in 1.5 mL tubes and ultrasonicated for 10 min. The concentration of DNA was measured. For DNA measurements, mixtures were centrifuged at 9500 g for 5 min, and supernatants were collected for DNA quantification. Isolated DAMPs and MAMPs were stored at −80 °C until further use.

## In vitro cytotoxicity assay
RAW 264.7 cells (cat. #TIB-71) were purchased from ATCC and were seeded into 96-well culture plates at a density of $2 \times 10^4$ cells/well and cultured until the cells were fully attached. Then the cells were treated with different concentrations of G3@SeHANs or soluble PAMAM-G3 for 24 h and were counted with a cell counting kit-8 (CCK-8).

## Extraction and quantification of cfDNA
Extraction of cfDNA from saliva or serum was performed with a DNeasy Blood & Tissue Kit (QIAGEN, Germany). Concentrations of cfDNA in

saliva, serum, DAMPs, and MAMPs were measured with a Quant-iT PicoGreen double-stranded DNA Assay Kit.

## Histone H3 pull-down

Pre-treatment of protein A/G magnetic beads (B23202, Bimake, China) was performed according to the manufacturer's instructions. Then, the serum sample was diluted two-fold with PBS and was added to protein A/G beads (100 µl beads for 150 µl serum) to pre-capture the natural IgG in the serum (2 h-incubation with mixing at 4 °C). After the beads were removed, the sample was further diluted four-fold with PBS to reduce the concentration of non-target protein. The ChIP-grade anti-histone H3 antibody was added to the diluted sample (1:200) and incubated with mixing at 4 °C overnight. The antibodies were captured using pretreated protein A/G magnetic beads (150 µl beads for 800 µl sample) and the beads were washed with 200 µl PBS 3 times to eliminate non-specific binding. The remaining serum sample after bead-removal and 3 × 200 µl PBS bead-washing buffer were combined and concentrated using 3 K ultrafiltration tubes (Amicon® Ultra 3Kdevice, Millipore-Merck, U.S.A). cfDNA extracted from the concentrated sample was labeled as "cfDNA after pull-down". cfDNA extracted from the magnetic beads was labeled as "cfDNA on beads". Both fractions of cfDNA were used for subsequent concentration and size measurement.

## DNA fragments size measurement

The size distribution of cfDNA fragments was obtained by using High Sensitivity DNA Chips and Reagents (5067-4626, Agilent Technologies, U.S.A.) on the 2100 Bioanalyzer system (Agilent Technologies, U.S.A.). The DNA concentration of the same sample was detected using Pico-Green as mentioned above. To approximate the concentration of cfDNA from different body fluids, serum-extracted cfDNA samples were diluted twice before testing.

## In vitro TLR3, TLR4, TLR8, and TLR9 activation assay

Stable hTLR3-, 4-, 8-, and 9-overexpressing HEK-Blue cells were purchased from InvivoGen (cat. #hkb-htlr3, hkb-htlr4, hkb-htlr8, and hkb-htlr9, respectively) (San Diego, CA, U.S.A.) and were initially propagated in DMEM with 10% (v/v) FBS and maintained in growth medium supplemented with selective antibiotics. Before treatment, certain numbers of HEK-Blue hTLR cells ($5 \times 10^4$ cells/well hTLR3, $2.5 \times 10^4$ cells/well hTLR4, $4 \times 10^4$ cells/well hTLR8, and $8 \times 10^4$ cells/well hTLR9 cells) were seeded and cultured in basal DMEM overnight in 96-well plates, then stimulated with the appropriate agonists (1 µg/mL low molecular weight poly(I:C) for hTLR3, 10 ng/mL ultrapure LPS for hTLR4, 0.5 µg/mL ORN06/LyoVec for hTLR8, and 1 µg/mL ODN BW006 for hTLR9). In the scavenger-treated groups, PAMAM-G3 (2 µg/mL), SeHANs (10 µg/mL), or G3@SeHANs (10 µg/mL) were added 30 min prior to adding the agonist. After 24 h, the activation of reporter cells was determined with QUANTI-Blue medium. One microliter of human saliva, 5 µL of human serum, DAMPs from PGKs (cfDNA concentration: 1 µg/mL), DAMPs from HGFs (cfDNA concentration: 1 µg/mL), DAMPs from mitochondria (cfDNA concentration: 500 ng/mL), MAMPs from *Porphyromonas gingivalis* (cfDNA concentration: 1 µg/mL), and MAMPs from and *Fusobacterium nucleatum* (cfDNA concentration: 1 µg/mL) were also prepared as agonists to test the function of cationic scavengers. Briefly, 50 µL supernatant from each well of the cell culture plate was transferred to 150 µL QUANTI-blue medium to test the corresponding secreted embryonic alkaline phosphatase (SEAP) activity, which was first loaded into the empty well and incubated at 37 °C for the reaction, and OD620 was measured.

## In vitro anti-inflammatory assays

RAW 264.7 cells were seeded and cultured in basal DMEM overnight at $2 \times 10^4$ cells per well in a 96-well plate. PAMAM-G3 (2 µg/mL) or G3@SeHANs (10 µg/mL) were added in a final volume of 200 µL 30 min prior to the addition of the agonist. ODN BW006 (1 µg/mL), 1 µL of human saliva, 5 µL of human serum, DAMPs from PGKs (cfDNA concentration: 1 µg/mL), DAMPs from HGFs (cfDNA concentration: 1 µg/mL), DAMPs from mitochondria (cfDNA concentration: 500 ng/mL), MAMPs from *Porphyromonas gingivalis* (cfDNA concentration: 1 µg/mL), and MAMPs from *Fusobacterium nucleatum* (cfDNA concentration: 1 µg/mL) were then added into the well. After incubation for 24 h, the supernatants were collected and TNF-α and IL-6 levels were measured using ELISA kits.

THP-1 cells (cat. #TIB-202) were purchased from ATCC and cultured in RPMI-1640 media supplemented with 10% FBS and selective antibiotics. Briefly, $8 \times 10^4$ cells were plated in 96-well plates in 200 µL RPMI media plus 25 ng/mL PMA for 48 h of treatment to induce differentiation to macrophages. For polarization, THP-1-derived macrophages were treated with PMA for 48 h, and periodontitis saliva was added during the final 18 h of treatment. PAMAM-G3 (2 µg/mL) or G3@SeHANs (10 µg/mL) were added in a final volume of 200 µL 30 min prior to adding saliva. After incubation, supernatants were collected and TNF-α, IL-6, TGF-β, IL-10, and BMP-2 levels were measured with ELISA kits.

## Animal model establishment and treatment

The animal study was approved by the Ethics Committee of West China Hospital of Stomatology, Sichuan University (WCHSIRB-D-2020-498). Male *BALB/C* mice (7-8 weeks old) were purchased from Dossy Experimental Animals Co., Ltd. (Chengdu, Sichuan, China) and maintained at the Experimental Animal Center of West China Second University Hospital, Sichuan University. Animals were fed standard food and water *ad libitum*, and light was provided according to natural circadian rhythms (temperature: 18–29 °C, humidity: 40–70%).

The murine experimental model of periodontitis was established by placing a ligature (Coated VICRYL Suture, 5-0; Ethicon | J&J Medical Devices, Somerville, New Jersey, U.S.A.) around the cervix of the maxillary second molar[39]. General anesthesia was administered by the intramuscular injection of Zoletil50 (50 mg/kg; Virbac, Carros, Grasse, France).

The circulating administration of the MSN-PEI (10 mg/kg, i.p.), which had been applied to sepsis[28], was performed 3 times a week, starting from the day of model establishment. The local administration of the materials (PAMAM-G3: 200 µg/mL; G3@SeHANs: 1 mg/mL) or PBS (in the control group) was accomplished by microinjection using a microsyringe (25 µL syringe with a 32-gauge needle; Hamilton Company, Reno, Nevada, U.S.A.) into gingival tissue at six sites (5 µL/site) around the ligature, including the mesiobuccal gingiva, distobuccal gingiva, mesiopalatal gingiva, distopalatal gingiva, and mesial and distal gingival papilla. To reduce the amount of anesthesia and local damage to the gingival tissue, microinjections were performed every three days (on days 0, 3, 6, 9, and 12); on experimental days without injection, the materials and PBS were noninvasively smeared on the gingiva using disposable microapplicators (M6500-SF purple; TPC Advanced Technology, Inc., City of Industry, California, U.S.A.).

To further determine the effect of pathological DNA on periodontitis, local injection of CpG (ODN2395, 100 µg/mL; InvivoGen, San Diego, California, U.S.A.) was performed in addition to ligature placement. The injection of CpG was performed as described above 30 min after each injection of materials or PBS.

## Animal body fluid and tissue sample collection

Saliva samples were collected before each injection after ligature placement and before sacrificing the animal, on days 3, 6, 9, 12, and 15. The saliva secreted by mice within 3 min after anesthesia was pipetted and collected. After centrifugation at 21,000 g for 10 min, the supernatant was collected and stored at −80 °C until further testing. For successful collecting the saliva, we suggested not using atropine during

anesthesia. Under this circumstance, the body position of mice needed to be paid attention for avoiding choking.

Animals were sacrificed 15 days after ligature placement. Oral microbiome samples were collected using an ultrafine polystyrene swab (25-800 1PD 50; Puritan Medical Products, Guilford, Maine, U.S.A.) according to a published protocol[79], and were stored at −80 °C until further testing. Blood samples were collected by eyeball extirpation under general anesthesia, coagulated at room temperature for 30 min, then centrifuged at $845\,g$ for 10 min to collect serum. Serum was stored at −80 °C until further testing.

After blood collection, animals were sacrificed by cervical dislocation. First, half of the maxilla on a random side was dissected and rinsed in cold PBS. The crown of the molars, maxillary bone, and buccal soft tissues were removed from samples, and only the gingival tissues and alveolar bone around the three molars were collected. The trimmed maxillary samples were temporarily placed on dry ice and stored at −80 °C for further testing. Then, the other side of the maxilla, heart, liver, spleen, lung, and kidney were dissected and fixed in 4% paraformaldehyde at 4 °C overnight. The fixed samples were rinsed with tap water for 6 h and were stored in 70% ethanol at 4 °C until further testing.

**Micro-CT reconstruction and bone resorption quantitative analyses.** Fixed maxillary samples were used first for micro-CT scanning (vivaCT80; SCANCO Medical, Brüttisellen, Switzerland). Scanning was performed at 145 mA and 55 kVp every 10 μm at high resolution. Further measurements of the vertical distance between the cementoenamel junction (CEJ) and alveolar bone crest (ABC) and three-dimensional reconstruction were performed with Mimics Research v19.0.0 (Materialise; Leuven, Belgium). The mesial and distal CEJ-ABC (μm) of the maxillary second molar was recorded as the mean of the numbers measured on five mesiodistal sectional planes. After three-dimensional reconstruction, a standard figure of the buccal and palatal aspect of the maxillary molars was obtained, on which the bone resorption area (mm²) was measured using Photoshop CC (Adobe; San Jose, California, USA). The bone resorption area was defined as the area enclosed by the continuous line of CEJ of the three molars and ABC.

### Histological analysis

After micro-CT scanning, maxillary samples were decalcified in 10% ethylene diamine tetra acetic acid-PBS solution at 4 °C for three weeks. Then, the decalcified maxilla and fixed organs were dehydrated, embedded in paraffin wax, and sectioned (4 μm). H&E staining and TRAP/ALP staining were performed using a staining kit.

For IF analysis, DCs were detected by an anti-CD11c Armenian hamster monoclonal antibody (1:100) and Alexa Fluor 647-conjugated goat anti-Armenian hamster IgG antibody (1:100). Macrophages were detected by an anti-F4/80 rat monoclonal antibody (1:50) and DyLight 488-conjugated AffiniPure goat anti-rat IgG (H + L) (1:100). In addition, markers of macrophage subpopulations were co-stained to identify the macrophage composition. M1-type and M2-type macrophages were stained by an anti-iNOS (1:400) and anti-arginase-1 (1:50) rabbit monoclonal antibody, respectively. 4′,6-diamidino-2-phenylindole (DAPI) was used to stain the cell nucleus. All IF-stained slices were mounted using ProLong Gold Antifade Mountant. Images were captured using an Upright Automated Fluorescence Microscope (BX63; Olympus, Tokyo, Japan).

For immunohistochemistry (IHC) analysis, the expression and location of TLR9 were detected by an anti-TLR9 rabbit polyclonal antibody (1:500). T cells were detected by an anti-CD3 rabbit polyclonal antibody (1:100). IHC was performed using an Anti-Rabbit HRP-DAB IHC Detection Kit. Images were captured using an Aperio ScanScope slide scanner (Leica Biosystems, Wetzlar, Germany), and relative quantitative analyses were performed by comparing the integral optical density (IOD) of the positive area using Image-Pro Plus v6.0.0 (Media Cybernetics; Rockville, Maryland, U.S.A.).

### Protein and RNA extraction from animal samples

Maxillary samples that were stored at −80 °C were homogenized using a high-speed tissue grinder (KZ-II; Servicebio, Wuhan, Hubei, China). Total RNA was extracted by using an RNApure Total RNA Fast Extraction Kit, and the quality and concentration of total RNA were assessed by a Nanodrop 2000 (Thermo Scientific, Waltham, Massachusetts, U.S.A.). Then, reverse transcription was performed using a RevertAid First Strand cDNA Synthesis Kit, and complementary DNA was stored at −20 °C until further testing.

Total protein used for ELISA was extracted using PathScan Sandwich ELISA Lysis Buffer. The concentration of total protein was measured by the BCA method. The total protein sample was stored at −80 °C until further testing.

### Cytokine concentration analysis

TNF-α and IL-6 levels in culture supernatants of RAW 264.7 cells and concentrations of TNF-α, IL-6, TGF-β, IL-10, and BMP-2 in culture supernatants of THP-1 cells were determined using ELISA kits.

### Quantitative real-time polymerase chain reaction assay

TRIzol reagent was used to extract total RNA, and 1 μg of total RNA was reverse-transcribed using an iScript One-Step RT-PCR Kit with SYBR Green. Quantitative polymerase chain reaction (PCR) was then performed using SYBR Green. Amplified transcripts were quantified using the comparative $Ct$ method.

### Biochemical parameter analysis

Alanine transaminase (ALT), aspartate transaminase (AST), blood urea nitrogen (BUN), total bilirubin (TBIL), phosphocreatine kinase (CK), and creatinine (CRE) were measured on a chemistry analyzer (Chemray-800, Rayto) with reagents and settings recommended by the manufacturer.

### Fluorescent labeling of cationic scavengers and the intracellular uptake of G3@SeHANs

FITC (150 μg) dissolved in DMSO (50 μL) was slowly added to PAMAM-G3 (3 mg) dispersed in sodium bicarbonate buffer (50 mM, pH 9.0, 0.4 mL) with vigorous stirring. The mixture was incubated for 12 h at 4 °C. FITC-labeled PAMAM-G3 (FITC-G3) was purified using an Amicon Ultra0.5 mL 10 K filter (Merck Millipore, Germany). FITC-G3 was used to coat SeHANs for further confocal laser scanning microscopy (CLSM) imaging.

RAW 264.7 cells were seeded onto a cover glass at a density of 10,000cells/cm². After 12 h, Cy5.5-labeled CpG (1 μg/mL) was added to cells with FITC-G3 (2 μg/mL) or FITC-G3@SeHANs (10 μg/mL) and incubated for 12 h. After 4 h, 8 h, and 12 h of treatment, the cells were washed with PBS, stained with LysoTracker Red DND-99 and DAPI, and mounted for CLSM.

### Flow cytometry analysis

THP-1 cells were treated with the method described in the in vitro anti-inflammatory assay section. Then, the cells were collected for flow cytometry analysis. Cells were washed with fluorescence-activated cell sorting (FACS) buffer (1x PBS plus 0.5% FBS) and blocked with Human TruStain FcX (Fc Receptor Blocking Solution). To identify DCs and monocytes, cells were stained with antibodies (1:20) against human CD14, CD83, CD86 and CD209 for 30 min at 4 °C. To identify macrophages and monocytes, cells were stained with antibodies against human CD14, CD36, CD68, and CD197. After staining, cells were washed and then resuspended in 1x DAPI in FACS buffer. Flow cytometry data were collected on an LSRFortessa or LSR II (BD Biosciences)

and analyzed using FlowJo software (Version 10.8.1, BD Biosciences). The exemplification of gating strategy was shown in Fig. S35.

### 16S RNA analysis

Genomic DNA from oral microbiome samples from mice was isolated and prepared for 16 S rRNA gene amplification (V1-V3 region) and sequencing at Majorbio (Shanghai, China) with a MiSeq 300 PE (Illumina MiSeq System) using primers (338 F 5'-ACTCCTACGGGAGG-CAGCAG-3'; 806 R 5'-GGACTACHVGGGTWTCTAAT-3'). The experiment was independently repeated four times. A total of four samples in each group were prepared ($n = 4$). Bioinformatics was performed with Mothur (v.1.43.0) and QIIME 2.0. Operational taxonomic unit (OTU) clustering was performed using USEARCH (v10). Sequences were aligned and taxonomically assigned with the Silva database. To avoid biases due to different sequencing depths, OTU tables were rarefied to the lowest number of sequences per sample. Analyses were performed on the I-Sanger Cloud Platform (http://www.i-sanger.com). For alpha diversity, the Shannon index, Chao index, Simpson index, and Ace index at the OTU level were calculated. For beta diversity, PCA at the OTU level was performed, and an analysis of similarities (ANOSIM) based on the Bray-Curtis distance was used to examine community differences. Average relative abundances of species are shown with Cicros plots and bar plots.

### Statistical analysis

The statistical analysis and most of the graphs production were accomplished by Prism 8 (GraphPad). Student's $t$-test was used to compare the mean value between the two groups. For multiple groups, a one-way analysis of variance and Tukey's multiple comparisons test was used. The correlation of clinical data was assessed by the spearman correlation coefficient (r), and the relevant scatter plots and heat maps were drawn by ggplot2 (3.3.5) and pheatmap package (1.0.12) on R (version 4.1.1).

### Reporting summary

Further information on research design is available in the Nature Research Reporting Summary linked to this article.

## Data availability

All the data generated in this study are provided in the Supplementary Information/Source Data file. All 16S rRNA genes sequences and genomes generated in the present study are available via Sequence Read Archive (SRA) using the individual accession numbers (PRJNA779377). Source data are provided in this paper.

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

## Acknowledgements

We are grateful to Chenghao Li, Xuanzhi Zhu, Chao-Jung Chu, Xiaoyu Wang, Chen Cui, and Wentao Jiang from West China Hospital of Stomatology, Sichuan University, Li Chen from Analytical and Test Center, Sichuan University, Xin Shi from Center of Stomatology, Tongji Hospital of Tongji Medical College, Huazhong University of Science and Technology, Jie Zhou and Huize Yan from Department of Biomedical Engineering, Columbia University, for expert technical assistance. This work was supported by National Natural Science Foundation of China (81991500, 81991502) granted to Q.C., National Key R & D Project (2018YFC1105701) and National Natural Science Foundation of China (31870960) granted to S.Z., National Natural Science Foundation of China (81901897) granted to Y.W., National Natural Science Foundation of China (81970944, 81991502) granted to L.Z., National Natural Science Foundation of China (81974147) granted to B.S., Research and Develop Program, West China Hospital of Stomatology, Sichuan University (RD-02-202107) grant to H. H., NIH AR073935 granted to K.W.L.

## Author contributions

H.H., W.P., and Y.W. contributed equally to this work. H.H., W.P., Y.W., H.K., D.S., B.H., TC.H., YH.L., C.H.Q. and J.S. contributed to the collection of experimental data. H.H., W.P., Y.W., H.K., D.S. and TC.H. analyzed the data. H.H., W.P., Y.W., H.K., D.S., TC.H., Q.C. and B.S. contributed to writing and revising the paper. S.Z., L.Z., and K.W.L. supervised the research.

## Competing interests

The authors declare no competing interests.

## Additional information

[1]State Key Laboratory of Oral Diseases and National Clinical Research Center for Oral Diseases, West China Hospital of Stomatology, Sichuan University, Chengdu, Sichuan 610041, China. [2]Department of Biomedical Engineering, Columbia University, New York 10027 NY, USA. [3]Department of Oral Maxillofacial Surgery, West China Hospital of Stomatology, Sichuan University, Chengdu, Sichuan 610041, China. [4]Department of Oral Medicine, West China Hospital of Stomatology, Sichuan University, Chengdu, Sichuan 610041, China. [5]Advanced Biomaterials and Tissue Engineering Center and Department of Biomedical Engineering, Huazhong University of Science and Technology, Wuhan, Hubei 430074, China. [6]Institute of Tissue Regeneration Engineering (ITREN), Dankook University, Cheonan 31116, the Republic of Korea. [7]Institutes for Life Sciences, School of Medicine, South China University of Technology, Guangzhou, Guangdong 510006, China. [8]Department of Orthopaedic Surgery, The Sixth Affiliated Hospital, Sun Yat-sen University and Guangdong Provincial Key Laboratory of Orthopaedics and Traumatology, Guangzhou, Guangdong 510000, China. [9]Department of Orthodontics and Pediatric Dentistry, School of Dentistry, The University of Michigan, Ann Arbor 48109 MI, USA. [10]Department of Periodontics, West China Hospital of Stomatology, Sichuan University, Chengdu, Sichuan 610041, China. [11]Department of Systems Biology, Columbia University Medical Center, New York 10032 NY, USA. [12]Present address: Stomatology Hospital, School of Stomatology, Zhejiang University School of Medicine, Clinical Research Center for Oral Diseases of Zhejiang Province, Key Laboratory of Oral Biomedical Research of Zhejiang Province, Cancer Center of Zhejiang University, Hangzhou, Zhejiang 310016, China. [13]Present address: School of Stomatology, Tongji Medical College, Huazhong University of Science and Technology, Wuhan, Hubei 430030, China. [14]These authors contributed equally: Hanyao Huang, Weiyi Pan, Yifan Wang. ✉e-mail: smzhang@mail.hust.edu.cn; jollyzldoc@163.com; kam.leong@columbia.edu

