## [Peer Review File · Nature Communications]

Reviewers' Comments:

Reviewer #1:

Remarks to the Author:

This study aims to show that cell-free DNA (cfDNA) is the culprit which causes periodontitis by triggering inflammatory response. The authors used HEK, RAW 264.7 and THP-1 cell line for their study. An inflammatory response in the form of TLR9 activation and cytokine secretion was detected in response to different sources of cfDNA, namely, saliva and serum of periodontitis patients, cfDNA isolated from saliva and serum of these patients, DAMPs (mtDNA and gDNA) isolated from gingival cells, PAMPs (CpG DNA) and MAMPs isolated from periodontitis bacteria. The authors show that the above inflammatory responses in vitro was abrogated by treatment with DNA scavengers, viz., nucleic acid-binding nanoparticles (NABNs) - G3@SeHANS and soluble PAMAM-G3. They also show that DNA scavengers alleviated periodontitis in a mouse model.

Major concern:

Although there is much current interest in cfDNA, for example in cancer diagnosis and therapy response, presence of cfDNA in circulation or in other body fluids is questionable. What exists in body fluids is cell-free chromatin since apoptotic cell death results in release of chromatin fragments and not of cfDNA [1]. The existence of cell-free chromatin in serum and / or plasma can be easily detected by ELISA [2], while the demonstration of cfDNA requires DNA to be extracted from plasma / serum using Proteinase-K treatment to remove proteins. In the present study the authors used the DNeasy Blood & Tissue Kit for isolation of cfDNA which has a Proteinase K treatment step and which removed the histones from chromatin to deliver cfDNA.

Multiple activities of cell-free chromatin in inducing inflammation have been reported [3-7], and the effects seen by the authors may well be due to presence of cell-free chromatin in saliva and serum. This supposition is consistent with the authors' observation of highly significant increase in TLR9 activation when they used saliva and / or serum of periodontitis patients, but not when DNA isolated from these body fluids or DAMPs were used.

The authors have used cfDNA scavenging nanoparticles which will also pull down chromatin (by binding to its DNA component), thereby abrogating the inflammatory activities of saliva and serum of periodontitis patients. Thus, again, the possibility cannot be excluded that it was chromatin present in the body fluids which was responsible for the authors' findings.

The authors should treat serum and saliva with anti-histone antibodies to exclude the possibility that the effects that they see are not due to chromatin particles present in these body fluids.

The authors may wish to consult the article by Marsman et al [8].

References

1. Van Nieuwenhuijze AEM, Van Lopik T, Smeenk RJT. Time between onset of apoptosis and release of nucleosomes from apoptotic cells: putative implications for systemic lupus erythematosus. *Ann Rheum Dis.* 2003 Jan; 62(1): 10–14. doi: 10.1136/ard.62.1.10
2. Holdenrieder S, Stieber P, Bodenmüller H, Fertig G, Fürst H, Schmeller N, et al. Nucleosomes in serum as a marker for cell death. *Clin Chem Lab Med.* 2001;39(7):596–605. DOI: 10.1515/CCLM.2001.095
3. Stefan Holdenrieder, Petra Stieber. Clinical use of circulating nucleosomes *Crit Rev Clin Lab Sci.* 2009;46(1):1-24. doi: 10.1080/10408360802485875.
4. Mittra I, Samant U, Sharma S, Raghuram G V., Saha T, Tidke P, et al. Cell-free chromatin from dying cancer cells integrate into genomes of bystander healthy cells to induce DNA damage and inflammation. *Cell Death Discov.* 2017 Dec 4;3(1):1–14. DOI: 10.1038/cddiscovery.2017.15
5. Mittra I, Pal K, Pancholi N, Tidke P, Siddiqui S, Rane B, et al. Cell-free chromatin particles released from dying host cells are global instigators of endotoxin sepsis in mice. Mukhopadhyay P, editor. *PLoS One.* 2020 Mar 4;15(3):e0229017. DOI: 10.1371/journal.pone.0229017
6. Chaudhary S, Mittra I. Cell-free chromatin: A newly described mediator of systemic inflammation. Vol. 44, *Journal of Biosciences.* Springer; 2019. p. 1–6. DOI: 10.1007/s12038-019-9849-7
7. Shabrish S, Mittra I. Cytokine Storm as a Cellular Response to dsDNA Breaks: A New Proposal. *Front. Immunol.*, 2021. Vol 12. DOI: 10.3389/fimmu.2021.622738
8. Marsman G, Zeerleder S, Luken BM. Extracellular histones, cell-free DNA, or nucleosomes: Differences in immunostimulation. Vol. 7, *Cell Death and Disease.* Nature Publishing Group; 2016. p. e2518. DOI: 10.1038/cddis.2016.410

Other comments:

1) The title of the manuscript is vague and does not reflect the main claims of the paper.

2) Abstract:

The authors claim to have demonstrated a correlation between cfDNA and periodontitis. However they do not actually demonstrate such a correlation, but only show that cfDNA levels are elevated in periodontitis patients. This has been reported previously in literature and are not new findings of this study (see references below).

References

1. White PC, Chicca IJ, Cooper PR, Milward MR, Chapple IL. Neutrophil Extracellular Traps in Periodontitis. J Dent Res. 2016 Jan;95(1):26-34. DOI: 10.3390/cells9061494

2. Boonyanit Thaweboon, Sukhumchawee Suwannagindra, Varunee Kerdvongbundit and Sroisiri Thaweboon. Using Absorbent Paper Strips for the Collection of Cell-Free DNA in Patients with Periodontal Diseases. 2019. Mater. Sci. Eng. 649 012010. DOI: 10.1088/1757-899X/649/1/012010

3. Sukhumchawee Suwannagindra, Boonyanit Thaweboon, Varunee Kerdvongbundit. Correlation between cell free DNA in gingival crevicular fluid and clinical periodontal parameters by using two collection techniques. M Dent J 2020; 40 (2) : 165-174

Consequently Line 257: "...we hypothesized that cfDNA may also be elevated in periodontitis..." needs to be modified.

3) NABNs in abstract are referred as G3@SeHANS in the manuscript. I would suggest that authors set a common nomenclature for the cfDNA scavengers throughout the manuscript.

4) Line 73: Add reference

5) The paragraph (lines 90-103) is confusing. It is not clear whether the authors are comparing G3@SeHANS with SeHANS or with PAMAM-G3? The objective of the study was to compare soluble form of PAMAM-G3 and G3@SeHANS. Why then were the immune modulatory effects of G3@SeHANS with bare SeHANS compared and discussed?

6) Line 210: On which tissue T cells and DCs were enumerated?

7) Line 32: In the abstract the authors have mentioned "Both cfDNA scavengers also regulated the mononuclear phagocyte system in a periodontitis environment, promoting the M1 over the M2 macrophage phenotype." However, in the results section, they say that scavenger treatment reduced M1 macrophages and increased M2 macrophages (Line 217-220). Finally, in the discussion authors say "NABPs and PAMAM-G3 did not reduce the M1 phenotype but did promote the M2 phenotype" (Line 345). Very confusing!

8) Line 265: "...scavenging cfDNA to block "abnormal" cfDNA-sensing pathways appears useful for treating uncontrolled localized inflammatory bone loss in periodontitis." Rather than abnormal, authors should use the term "hyperactive" or "over activated".

9) Materials and Methods section

- Statistical analysis paragraph is missing.
- The authors should mention the genes evaluated by Real time PCR and also details of house-keeping gene used.

-Dr. Snehal Shabrish

Reviewer #2:

Remarks to the Author:

The authors report on a potential therapeutic approach for periodontal disease which is based on the hypothesis that presence of cell-free DNA in the periodontal tissues/exudates is correlated with periodontitis through a mechanism of TLR9 interactions of cell-free DNA and removal of the cell-free DNA will ameliorate the disease. This is an interesting paper with potentially innovative approach in the treatment of periodontitis and potentially related systemic diseases, however there are several unclear statements in the use of these nanoparticles as well as the results of the treatment of mouse model of periodontitis and require major revisions. Specifically:

1. The references (Ref 4 and 5) used are not specifically supporting the authors claims related to periodontal disease; such as the reference for involvement of cell-free DNA in periodontitis; "One type of molecular pattern that contributes to periodontitis is cell-free DNA (cfDNA)". The paper does not refer to periodontitis at all. Regarding reference 4, there are periodontitis specific references describing the involvement of pattern recognition receptors and periodontal

inflammation (i.e., Wallet SM, Puri V, Gibson FC, 2018).

2. Similarly, references 15 and 16 do not refer to periodontal disease as the authors stated.

"cfDNA is critical to chronic inflammation related to bone loss in periodontitis". These references refer to rheumatoid arthritis.

3. Once again, the authors state that "cfDNA/TLR9 interactions play role in the periodontitis immune response" without any reference.

4. The aim of this study therefore should be revised to determine the role of cfDNA in periodontal immune system and determine the TLR9 interactions in alveolar bone loss. The inhibition of this interaction or removal of cfDNA by using the nanoparticles should be used to prove the mechanism, not as the main purpose of this study.

5. Presentation of the study results is unorganized and confusing. Based on the concerns above, the aim should be to demonstrate the role of cfDNA in periodontitis and potential interaction with TLR9. Thus, the results should first demonstrate this interaction. In vitro experiments with HEK TLRs should be separated from those RAW 264.7 macrophages.

6. The authors state that G3@SeHANS caused greater inhibition than bare SeHANS, but interestingly the higher dose of G3@SeHAN resulted in a similar inhibition with 5 fold less dose of (10 µg/mL vs 2 µg/mL) soluble PAMAM-G3. Please explain.

7. Page 4, Line 111-129 should be presented as the first experiment and finding –aim should be revised.

8. Once again, the results section is difficult to follow—confusing statements and not presented in a logical order. Additional subheadings are strongly suggested.

9. Please clarify what "related phenotypes" was referred to in the following sentence? "The ligature was placed in a circle around the second molar to obtain significant bone loss and related phenotypes". Also, what does "significant bone loss" refer to? What is significant bone loss? Quantification should be given.

10. The following statements are not clear: PBS, PAMAM-G3, or G3@SeHANS were injected on both sides of mice (what are the locations of the injections?) into six sites around the ligature on days 0, 3, 6, 9, and 12. Samples (what samples?) were collected on day 15. Treatments were also smeared onto the gum (how and why was this performed?) around the ligature daily except on injection days. Please explain the logic and mechanism of topical application and impact on gingiva.

11. The authors indicate that saliva was collected after anesthesia—which is normally contradictory to saliva secretion, anesthesia cause dry mouth in mice—was saliva secretion induced by any pharmacological agent (pilocarpine, for example)? How much saliva was obtained?

12. Were ligatures kept during the 2 weeks? Were there any loss of ligatures and replacement during this period?

13. Why the plaque samples were collected with swaps instead of by ligatures?

14. Maxilla was first used for microCT then for histology, but also for protein extraction. Please clarify, which maxillary samples were used for this.

15. 16S RNA shows no impact on microbiome in total, need to add to discussion the explanation of these results.

16. Fig S18—the differences hard to be seen. Please explain.

17. "cfDNA and TNF-α levels in saliva and serum were significantly higher in periodontitis model mice than in normal mice; these proinflammatory cytokine levels (there is only one cytokine measured) were inhibited by treatment with...".

18. Were treatments initiated after 2 weeks of periodontitis induction? What do days 0, 3, 6, 9, and 12 mean?

19. MicroCT images (Fig. 3I-J) do not show much bone loss reduction as stated by the authors. Please explain.

20. Histological images show similar degree of inflammation, it is hard to see the differences.

21. The ligature model used is more of a traumatic model (5-0 sutures) than bacterial inflammatory model.

22. The relevance of these periodontitis models to human chronic periodontitis should be discussed in the discussion especially those aggressive and acute bone loss models.

23. It is interesting that the most benefit was shown by G3@SeHAN was extracellularly localized, as it is know that cfDNA was more involved in cytoplasm. Please explain.

24. English grammar use and writing of entire manuscript (organizational) should be reviewed and reorganized.

Reviewer #3:

Remarks to the Author:

The manuscript entitled " Cationic Nanostructures for the Treatment of Inflammatory Bone Loss " developed a bone-mimicking 28 selenium-doped hydroxyapatite nanoparticles with cationic polyamidoamine dendrimers 29 (PAMAM-G3), and compared the activities of these NABNs with those of soluble PAMAM-G3 30 polymers for treatment of inflammatory bone loss. Generally speaking, the introduction is insufficient, and the authors should provide more evidence to support the formation of the bone-mimicking nanoparticles. Therefore, I cannot accept it at the current stage:

1. The introduction in the present study does not make the significance and impact of the work clear.
2. What amount of Se was doped into HA? How to determine it?
3. The authors should carefully clarify why they synthesize nanoparticles are bone-mimicking in terms of chemistry, structure or biological property. Because the apatite in bone is complicated and has different configurations, it is recommended to compare their synthesized "bone-mimicking" nanoparticles with natural apatite nanoparticles in bone, so that to demonstrate that their fabricated nanoparticle are bone-mimicking.
4. The pictures should be further improved, and the pictures should be aligned with each other. For example, Figure 1a and Figure 1b are not aligned.
5. Between each picture, the font format and symbol format should be consistent within the same picture. Please adjust them.

Response to Reviewers' comments

We would like to thank the reviewers for their constructive comments.

We have complied with all the editorial requests.

1. Editorial policy checklist and reporting summary have been finished.
2. Data Availability section is included.
3. All 16S rRNA genes sequences and genomes generated in the present study are available via Sequence Read Archive (SRA) using the individual accession numbers (PRJNA779377).
4. 'Source Data' file is updated and submitted.
5. We have revised all our bar graphs as requested.

Please find our detailed response in a point-to-point manner as listed below. Corresponding changes have been made in the manuscript and Supplementary information as indicated.

Reviewer #1 (Remarks to the Author):

This study aims to show that cell-free DNA (cfDNA) is the culprit which causes periodontitis by triggering inflammatory response. The authors used HEK, RAW 264.7 and THP-1 cell line for their study. An inflammatory response in the form of TLR9 activation and cytokine secretion was detected in response to different sources of cfDNA, namely, saliva and serum of periodontitis patients, cfDNA isolated from saliva and serum of these patients, DAMPs (mtDNA and gDNA) isolated from gingival cells, PAMPs (CpG DNA) and MAMPs isolated from periodontitis bacteria.

The authors show that the above inflammatory responses in vitro was abrogated by treatment with DNA scavengers, viz., nucleic acid-binding nanoparticles (NABNs) - G3@SeHANS and soluble PAMAM-G3. They also show that DNA scavengers alleviated periodontitis in a mouse model.

Major concern:

Although there is much current interest in cfDNA, for example in cancer diagnosis and therapy response, presence of cfDNA in circulation or in other body fluids is questionable. What exists in body fluids is cell-free chromatin since apoptotic cell death results in release of chromatin fragments and not of cfDNA [1]. The existence of cell-free chromatin in serum and / or plasma can be easily detected by ELISA [2], while the demonstration of cfDNA requires DNA to be extracted from plasma / serum using Proteinase-K treatment to remove proteins. In the present study the authors used the DNeasy Blood & Tissue Kit for isolation of cfDNA which has a Proteinase K treatment step and which removed the histones from chromatin to deliver cfDNA.

Multiple activities of cell-free chromatin in inducing inflammation have been reported [3-7], and the effects seen by the authors may well be due to presence of cell-free chromatin in saliva and serum. This supposition is consistent with the authors' observation of highly

significant increase in TLR9 activation when they used saliva and / or serum of periodontitis patients, but not when DNA isolated from these body fluids or DAMPs were used.

The authors have used cfDNA scavenging nanoparticles which will also pull down chromatin (by binding to its DNA component), thereby abrogating the inflammatory activities of saliva and serum of periodontitis patients. Thus, again, the possibility cannot be excluded that it was chromatin present in the body fluids which was responsible for the authors' findings.

The authors should treat serum and saliva with anti-histone antibodies to exclude the possibility that the effects that they see are not due to chromatin particles present in these body fluids.

The authors may wish to consult the article by Marsman et al [8].

References

1. Van Nieuwenhuijze AEM, Van Lopik T, Smeenk RJT. Time between onset of apoptosis and release of nucleosomes from apoptotic cells: putative implications for systemic lupus erythematosus. *Ann Rheum Dis.* 2003 Jan; 62(1): 10–14. doi: 10.1136/ard.62.1.10
2. Holdenrieder S, Stieber P, Bodenmüller H, Fertig G, Fürst H, Schmeller N, et al. Nucleosomes in serum as a marker for cell death. *Clin Chem Lab Med.* 2001;39(7):596–605. DOI: 10.1515/CCLM.2001.095
3. Stefan Holdenrieder, Petra Stieber. Clinical use of circulating nucleosomes *Crit Rev Clin Lab Sci.* 2009;46(1):1-24. doi: 10.1080/10408360802485875.
4. Mitra I, Samant U, Sharma S, Raghuram G V., Saha T, Tidke P, et al. Cell-free chromatin from dying cancer cells integrate into genomes of bystander healthy cells to induce DNA damage and inflammation. *Cell Death Discov.* 2017 Dec 4;3(1):1–14. DOI: 10.1038/cddiscovery.2017.15
5. Mitra I, Pal K, Pancholi N, Tidke P, Siddiqui S, Rane B, et al. Cell-free chromatin particles released from dying host cells are global instigators of endotoxin sepsis in mice. Mukhopadhyay P, editor. *PLoS One.* 2020 Mar 4;15(3):e0229017. DOI: 10.1371/journal.pone.0229017
6. Chaudhary S, Mitra I. Cell-free chromatin: A newly described mediator of systemic inflammation. Vol. 44, *Journal of Biosciences.* Springer; 2019. p. 1–6. DOI: 10.1007/s12038-019-9849-7
7. Shabrish S, Mitra I. Cytokine Storm as a Cellular Response to dsDNA Breaks: A New Proposal. *Front. Immunol.*, 2021. Vol 12. DOI: 10.3389/fimmu.2021.622738
8. Marsman G, Zeerleder S, Luken BM. Extracellular histones, cell-free DNA, or nucleosomes: Differences in immunostimulation. Vol. 7, *Cell Death and Disease.* Nature Publishing Group; 2016. p. e2518. DOI: 10.1038/cddis.2016.410

Reply:

The point is well-taken. We appreciate the comment, as it involves some ambiguous and still controversial definitions of the terminology related to cfDNA. The process of understanding this issue has greatly helped us clarify the concept of cfDNA in the manuscript.

The question posed by the reviewer did raise our curiosity about the proportion of pure DNA (protein-free) and DNA-protein complexes within cfDNA in body fluids. Indeed, as described in the references provided by the reviewer, studies on serum cfDNA suggest that most of the cfDNA is in the form of cell-free nucleosomes, and the size characteristics of cfDNA suggest similar conclusions (Integer multiple of about 170bp). However, we did not find studies addressing salivary cfDNA composition and the nucleosome content in saliva. Thus, we performed pull-down experiments against histone H3 as suggested by the reviewer. Unfortunately, we encountered the following difficulties and problems.

First of all, we needed to find a way to determine the success of pull-down of histone H3. To achieve this goal, we have to quantify histone H3 in the body fluids before and after pull down, and on the magnetic beads used in the pull-down. As described by the reviewers, most studies chose to quantify plasma histone h3 using ELISA. Unfortunately, using the total H3 ELISA kit (53110, Active Motif, U.S.A) with the highest detection sensitivity we could acquire, we could not detect the serum and saliva histone H3 concentrations within the detection range. With the prolonged antigen capture time (4°C overnight, RT for 1h is recommended) and prolonged color development time (up to 30min, 3-5min is recommended), some positive signals could be observed in both serum and saliva samples when compared with blank control (Figure below), and no difference was found between the two body fluids. However, these values are not covered by the standard curve and may not be accurate, and hence could not be relied upon to assess the efficiency of pull down.

A. The absorbance unit obtained when performing total H3 ELISA after prolonged color development time between serum and saliva samples from periodontitis patients when compared with blank (PBS).

B. The DNA concentration of the 6 samples used in Fig. A.

Also, we have carefully read the reference provided by the reviewer¹, which explored an assay to assess nucleosome levels in fluids, achieved by modifying an ELISA kit that detects cell death. The antibodies used in this kit were complex, consist of anti-histones (H1, H2, H3 and H4) and anti-DNA (ssDNA and dsDNA) antibodies. By using these antibodies, this kit in fact detects a collection of molecules encompassing cell-free pure DNA (protein-free), nucleosomes and free histones, but it is not a specific assay for histone-DNA complexes.

Also, the authors modified the method by using a long color development time (30 min) and compared the differences in absorbance unit between groups, similar to our attempt above. The quantification of histone could not be accomplished by this method as well.

In fact, other studies have also shown that the level of circulating histone H3 was too low to be detected in healthy individuals both for human and mouse²⁻⁴, which is consistent with our results. Very high levels were detected only in patients groups (mostly severe systemic diseases such as lung injury, trauma and sepsis). Unfortunately, H3 levels in serum and saliva of patients with periodontitis were also below the detection range of the kit. This is consistent with the difference of cfDNA level (2-3 fold) in body fluids between periodontitis patients and healthy volunteers. Clearly, periodontitis, as a local inflammatory disease, can affect circulating cfDNA, but the effect is not as high as those in severe systemic inflammatory conditions such as sepsis or rheumatoid arthritis.

However, given that DNA on nucleosomes and free DNA are not identical in terms of size pattern, we made a new round of attempts. We tried to use a ChIP grade anti-histone H3 antibody (ab1791, Abcam, UK) and protein A/G magnetic beads for IP (B23202, Bimake, China) for nucleosome pulldown, and then detected the difference in concentration and size pattern of the DNA on the beads and supernatants respectively, and compared them with the original body fluid. Unfortunately, no DNA was detected on the beads. We speculated that this might be due to the high albumin in the serum that affected the recognition of histone H3 by the capture antibody, or the high globulin content that affected the binding of the capture antibody by the beads. Therefore, we next made attempts to remove albumin and globulin from the serum samples.

We first tried an Albumin and IgG Erasin Kit (C500063, Sangon Biotech, China). In these experiments, protein A sefinose incubation was applied to remove globulin and cryo-ethanol method was used to remove albumin. The cryoethanol method is an industrial way to prepare human serum albumin, and the purity of the extracted albumin is good. However, the efficiency of albumin removal is limited. Besides, residual proteins would form an extremely dense precipitate after the final step of prolonged high speed centrifugation (20,000g for 20min), which was difficult to dissolve using mild buffers and conditions. The samples processed in this way did not meet the requirements of the subsequent pull-down experiments. We then tried to replace the albumin removal method with Affi-Gel® Blue Gel (153-7301, Bio-Rad, U.S.A.). These affinity dye gel microspheres could adsorb albumin at low salt concentrations for a short time and with high efficiency. After adjusting the experimental conditions, we successfully achieved the removal of about 60% of the proteins from serum while ensuring the retention of more than 90% of the DNA (Figure below).

- A.** The protein concentration before and after albumin and immunoglobulin removal obtained by BCA (PC0020, Solarbio, China); albumin would be released from the blue gel in high salt buffer (1.4M NaCl in 20mM phosphate buffer, pH = 7.1).
- B.** The DNA concentration of the samples assessed in Fig. A.

The process of albumin removal (4x dilution for reducing ionic strength, 2x for washing) and post-IP magnetic bead washing (3X) would result in serum samples being diluted up to 24-fold, and we also use ultrafiltration tubes (Amicon® Ultra 3Kdevice, Millipore-Merck, U.S.A) for protein and DNA enrichment. However, unfortunately, even after performing the above treatment followed by pulldown of H3, we still did not detect the presence of DNA on the beads. In addition to this, we also tried replacing fresh samples without frozen storage, adding buffers to stabilize the pH of the body fluid, and adding protease inhibitors, all of which were also unsuccessful.

Based on the above attempts, we regret to say that we were unable to successfully implement the pull-down experiments suggested by the reviewer under the existing reagents, kits and conditions. However, in this process, we examined the size characteristics of cfDNA fragments from serum and saliva as mentioned above, and the results were meaningful. Not surprisingly, the serum-extracted cfDNA possessed specific distribution pattern of DNA fragment size, which represented as integer multiples of base pairs in single nucleosome. This pattern suggested that cfDNA within the serum might be derived from endogenous genomic DNA released by dead host cell, and present as extracellular nucleosomes. Interestingly, however, there is no such distinct distribution pattern for saliva cfDNA. This indicated, to some extent, that saliva had a larger proportion of exogenous origin (bacterial) of cfDNA. The capacity of activating TLR9 by DNA of such origin is stronger than genomic DNA due to the existence of unmethylated CpG motif, which might be the reason why DNA extracted from saliva and serum did not show the same activation capacity on TLR9 reporter cells. These results and discussion have been added to the Results and Discussion section.

However, there not-so-successful attempts above did not deter us from taking this issue seriously. In fact, there are indeed many details worth discussing in this issue.

As stated by the reviewers, cfDNA has increasingly come into our view due to the rise of the concept of tumor liquid biopsy. Most of the research on cfDNA has been done by oncology scientists, often adopting cfDNA as an expression for circulating-tumor DNA (ctDNA) in many settings. In those studies, it does not matter in what form the DNA is present in the circulatory system, but more importantly the sequence and epigenetic information carried by the DNA fragments. So, in this context, the concept of cfDNA is more often considered as the collection of all the molecules that is outside the cell (cell-free) and contains DNA. This includes: truly free DNA fragments without association with proteins, and complexes of DNA and proteins (including histone and other DNA binding protein). It is true that such a nomenclature (cfDNA for all these molecules) is not appropriate from a biochemical point of view, since protein-DNA complexes cannot be referred to as DNA, but unfortunately there is no universally accepted standard to describe this collection of substances. Therefore, we have also chosen to use the term cfDNA, which has the widest acceptance, to refer to all cell-free DNA-containing molecules. Similar choices were also made by articles discussing cfDNA and other inflammatory diseases in recent years⁵⁻⁸. Besides, DNA without bound protein is generally referred to as “free DNA”⁹.

In addition, studies and reviews on the differences in capacity of immunostimulation of various components of cfDNA suggest very interesting information. It is currently believed that free nucleosomes, free DNA and other DNA-protein complexes all have the ability to activate TLR9, but **histone alone activates TLR2/4**. These result suggests that among the components of cfDNA, it is the DNA itself that actually activates TLR9¹⁰. Related studies have also shown that the DNA backbone exposed by folding the DNA over the protein may be the effective component of the complex to activate TLR9¹¹.

Taken together, all the above findings suggest an interesting conclusion. It is the DNA itself that really matters in cfDNA, whether from the perspective of oncology or immunology. Therefore, until a more precise and accepted term becomes available, we believe that the use of cfDNA to refer to all DNA-containing substances in immune response-related studies is acceptable. However, we very much appreciate the critique of the reviewer. We included a brief discussion on the concept of cfDNA in the Discussion section to make sure that we are not over-simplifying the concept and misleading the field.

Based on the previously mentioned information, as for current commercially available kits for DNA extraction or specialized cfDNA extraction, samples need to be treated with proteinase K in the first step to adequately release and extract DNA. During this revision, we also briefly explored the effect of not adding proteinase K during the DNA extraction process on the DNA yields. The results showed that despite little change in saliva, not treating serum with proteinase K would greatly reduce the yield of DNA. This may be related to the excessive protein content of the serum.

We next discuss the result that the body fluid as a whole has a strong activation effect on TLR9 Reporter HEK293 cells, but the activation ability decreased after DNA extraction. First, the established principle of TLR9 Reporter HEK293 Cells is to overexpress TLR9 gene within HEK293 Cells while transfected with a reporter gene SEAP that can be up-regulated by NF- κ B activation. On the basis of this, levels of SEAP in the supernatant can be easily determined by adding HEK-Blue™ Detection substrate.

This information suggests that the cells would be very sensitive to TLR9 agonists, but any activation of the NF- κ B pathway by other stimuli could also up-regulated the reporter gene SEAP expression. This deduction can also be supported by the information provided on the official website of Invivogen below:

TLR/NOD Induction Response of HEK-Blue™ hTLR9 cells to TLR and NOD agonists

<https://www.invivogen.com/hek-blue-hltr9>

The contents of body fluids are very complex, and any potentially unknown TLR9 agonist or NF- κ B activator (e.g. TNF α) can activate the reporter gene transcription in TLR9 Reporter HEK293 cells. Therefore, the significant activation of TLR9 Reporter Cells by body fluids might not necessarily arise from the DNA protein complexes disrupted during DNA extraction. The potential causes of the decrease of activation may be very complex, which might be the reason why, in experiments with body fluid samples, only partial activation of TLR9 reporter cells could be inhibited by G3@SeHANs. Next, we further examined the differences in the immunostimulation ability on TLR9 between various types of DNA. The results also showed that the TLR-activating ability of certain DNAs (saliva DNA, CpG, mitochondrial DNA) could be mostly inhibited by G3@SeHANs.

Other comments:

1) The title of the manuscript is vague and does not reflect the main claims of the paper.

Reply:

Thanks for your kind suggestion. We have revised the title as “Nanoparticulate Cell-free DNA Scavenger for Treating Inflammatory Bone Loss in Periodontitis”, which pointed out

the cfDNA scavenging and periodontitis.

2) Abstract:

The authors claim to have demonstrated a correlation between cfDNA and periodontitis. However they do not actually demonstrate such a correlation, but only show that cfDNA levels are elevated in periodontitis patients. This has been reported previously in literature and are not new findings of this study (see references below).

References

1. White PC, Chicca IJ, Cooper PR, Milward MR, Chapple IL. Neutrophil Extracellular Traps in Periodontitis. *J Dent Res*. 2016 Jan;95(1):26-34. DOI: 10.3390/cells9061494
2. Boonyanit Thaweboon, Sukhumchawee Suwannagindra, Varunee Kerdvongbudit and Sroisiri Thaweboon. Using Absorbent Paper Strips for the Collection of Cell-Free DNA in Patients with Periodontal Diseases. 2019. *Mater. Sci. Eng.* 649 012010. DOI: 10.1088/1757-899X/649/1/012010
3. Sukhumchawee Suwannagindra, Boonyanit Thaweboon, Varunee Kerdvongbudit. Correlation between cell free DNA in gingival crevicular fluid and clinical periodontal parameters by using two collection techniques. *M Dent J* 2020; 40 (2) : 165-174
Consequently Line 257: “.we hypothesized that cfDNA may also be elevated in periodontitis...” needs to be modified.

Reply:

Based on the reviewers' suggestions, we further explored the correlation between cfDNA concentration and the severity of periodontal inflammation during this revision. To achieve this goal, we performed two tasks:

1. Supplemented the integrated information on the periodontal clinical examination of the participants.

For the 25 participants included in the original data, 10 healthy volunteers and 10 periodontitis patients were screened for further analysis since the complete record of demographic data and periodontal clinical examination could be obtained.

Among the several periodontal clinical examination parameters, we selected the most representative ones, including dental plaque index (PLI), bleeding index (BI), and probing depth (PD), to reflect the intensity of local pathogen irritation, the degree of periodontal soft tissue inflammatory infiltration, and the degree of periodontal hard tissue destruction, respectively.

2. Included a new group of patients, namely gingivitis, as a transitional state between healthy and chronic periodontitis.

To reflect the progression of periodontitis pathogenesis, we chose to include the gingivitis patients, an antecedent stage of most periodontitis, as an additional group of clinical samples.

By integrating the above data, we found that only the concentration of cfDNA in saliva but not serum was increased in the gingivitis group.

Our results also showed a correlation of cfDNA with the severity of periodontal local inflammation in both saliva and serum. Interestingly, the correlation was stronger in saliva than in serum. We have revised the text in the Result section 1 “cfDNA levels in saliva and serum are correlated with the severity of periodontal inflammation”, Fig. 1 and Fig. S1-4 to demonstrate the correlation.

3) NABNs in abstract are referred as G3@SeHANs in the manuscript. I would suggest that authors set a common nomenclature for the cfDNA scavengers throughout the manuscript.

Reply:

Thank you for your suggestion. We have clearly defined G3@SeHANs as the nucleic acid-binding nanoparticles specifically for periodontitis and set G3@SeHANs as the common nomenclature in the revised manuscript.

4) Line 73: Add reference

Reply:

Thanks. We have added the reference¹².

5) The paragraph (lines 90-103) is confusing. It is not clear whether the authors are comparing G3@SeHANs with SeHANs or with PAMAM-G3? The objective of the study was to compare soluble form of PAMAM-G3 and G3@SeHANs. Why then were the immune modulatory effects of G3@SeHans with bare SeHans compared and discussed?

Reply:

Thanks for pointing this out. The comparison between SeHANs and G3@SeHANs was aimed to demonstrate the scavenging capacity after the coating of PAMAM-G3. It was to prove that cationic polymer coating endowed SeHANs with significantly enhanced immune modulatory effects. We moved this part to the New Results section, “Cationic G3@SeHANs can function as DNA scavenger”, which demonstrated the fabrication of cfDNA-scavenging G3@SeHANs.

6) Line 210: On which tissue T cells and DCs were enumerated?

Reply:

It was the periodontal tissue. We have revised it in the respective parts about T cells and DC in the Results section “G3@SeHANs regulate the mononuclear phagocyte system”.

7) Line 32: In the abstract the authors have mentioned “Both cfDNA scavengers also regulated the mononuclear phagocyte system in a periodontitis environment, promoting the M1 over the M2 macrophage phenotype.” However, in the results section, they say that scavenger treatment reduced M1 macrophages and increased M2 macrophages (Line 217-220). Finally, in the discussion authors say “NABPs and PAMAM-G3 did not reduce the M1 phenotype but did promote the M2 phenotype” (Line 345). Very confusing!

Reply:

Sorry for these typos that led to misunderstanding. For the abstract, it should be promoting the M2 over the M1 macrophage phenotype. As shown by flow cytometry, the scavenger could increase monocytes, M0 and M2 macrophages, when compared with the situation caused by periodontitis saliva. Periodontitis saliva would increase the M1 when compared to healthy saliva. By ELISA, for the M1 phenotype, levels of TNF- α and IL-6 increased following exposure to periodontitis saliva, but this increase was significantly reduced by treatment with G3@SeHANs and PAMAM-G3, which also confirmed the M1 phenotype could be induced by periodontitis saliva. Together, scavenger treatment reduced M1 macrophages and increased M2 macrophages. It have been clearly defined in the revised manuscript.

8) Line 265: “...scavenging cfDNA to block “abnormal” cfDNA-sensing pathways appears useful for treating uncontrolled localized inflammatory bone loss in periodontitis.” Rather than abnormal, authors should use the term “hyperactive” or “over activated”.

Reply:

Thanks for your suggestion. We have changed “abnormal” to “hyperactive”.

9) Materials and Methods section

- Statistical analysis paragraph is missing.
- The authors should mention the genes evaluated by Real time PCR and also details of house-keeping gene used.

Reply:

Thanks for pointing these out. We have added the statistical analysis paragraph at the end of revised Materials and Methods section. The genes and house-keeping genes have been included in the new Tab. S3.

Reviewer #2 (Remarks to the Author):

The authors report on a potential therapeutic approach for periodontal disease which is based on the hypothesis that presence of cell-free DNA in the periodontal tissues/exudates is correlated with periodontitis through a mechanism of TLR9 interactions of cell-free DNA and removal of the cell-free DNA will ameliorate the disease. This is an interesting paper with potentially innovative approach in the treatment of periodontitis and potentially related

systemic diseases, however there are several unclear statements in the use of these nanoparticles as well as the results of the treatment of mouse model of periodontitis and require major revisions. Specifically:

1. The references (Ref 4 and 5) used are not specifically supporting the authors claims related to periodontal disease; such as the reference for involvement of cell-free DNA in periodontitis; “One type of molecular pattern that contributes to periodontitis is cell-free DNA (cfDNA)”. The paper does not refer to periodontitis at all. Regarding reference 4, there are periodontitis specific references describing the involvement of pattern recognition receptors and periodontal inflammation (i.e., Wallet SM, Puri V, Gibson FC, 2018).

Reply:

Thanks for your kind suggestion. We have revised this sentence as “Over-activated TLR signaling can lead to periodontitis” and added your recommended reference.

2. Similarly, references 15 and 16 do not refer to periodontal disease as the authors stated. “cfDNA is critical to chronic inflammation related to bone loss in periodontitis”. These references refer to rheumatoid arthritis.

Reply:

Thanks. We have revised the writing to “... cfDNA is critical to chronic inflammatory bone loss in rheumatoid arthritis”.

3. Once again, the authors state that “cfDNA/TLR9 interactions play role in the periodontitis immune response” without any reference.

Reply:

Thanks for pointing this out. The reason we did not cite any reference is because there is none. The literature only reported the observation of TLR9 activation in periodontitis, and increase of DNA levels in periodontitis. We have revised the text on the potential role of TLR9 role in periodontitis and added the related reference in the Result section “Cationic G3@SeHANS can function as DNA scavenger”.

4. The aim of this study therefore should be revised to determine the role of cfDNA in periodontal immune system and determine the TLR9 interactions in alveolar bone loss. The inhibition of this interaction or removal of cfDNA by using the nanoparticles should be used to prove the mechanism, not as the main purpose of this study.

Reply:

Thanks for your suggestion. We have rearranged the structure of our manuscript. First, we studied the correlation of cfDNA with the progression of periodontitis. Specific correlation between the cfDNA levels in body fluids and periodontitis, including different parameters for

determining the severity of periodontal inflammation, had been demonstrated. Then based on this finding, we fabricated the periodontitis-special cfDNA scavenger, tested its DNA-scavenging efficiency and inhibitory effect on TLR signaling. The next topic was that whether cellular TLR9 signaling could be inhibited by scavenging periodontitis-derived cfDNA, and whether the nanoparticles could inhibit the TLR9 activation. Meanwhile, the cfDNA-scavenging mechanisms were demonstrated for further explaining the possible differences in the animal study. Thus, through cfDNA-scavenging strategy, which can be proven by the decrease of cfDNA, periodontitis-related inflammatory bone loss was inhibited while TLR9 signaling was reduced. Following this structure, **the aim of this study therefore has been revised to determine the role of cfDNA in periodontal immune system, and prove that cfDNA-scavenging would affect TLR9 signaling, which in turn can alleviate alveolar bone loss.**

Accordingly, the introduction was rearranged to focus on introducing the role of TLR9 in periodontitis, and the role of cfDNA in periodontitis. We also stated that DNA sensing signaling can be blocked by application of cfDNA scavenger, and in this study, can help prove the role of cfDNA in proinflammatory immune response in periodontitis.

5. Presentation of the study results is unorganized and confusing. Based on the concerns above, the aim should be to demonstrate the role of cfDNA in periodontitis and potential interaction with TLR9. Thus, the results should first demonstrate this interaction. In vitro experiments with HEK TLRs should be separated from those RAW 264.7 macrophages.

Reply:

Thanks for your kind suggestion. We rearranged our results to separate HEK TLR9 and RAW 264.7 macrophages, where the results of *in vitro* experiments with macrophages were set in the last paragraph of the New Results section “Cellular TLR9 signaling can be inhibited by scavenging periodontitis-derived cfDNA”.

6. The authors state that G3@SeHANs caused greater inhibition than bare SeHANs, but interestingly the higher dose of G3@SeHAN resulted in a similar inhibition with 5 fold less dose of (10 µg/mL vs 2 µg/mL) soluble PAMAM-G3. Please explain.

Reply:

According to the TG analysis, the mass percentage of PAMAM-G3 coating on the surface of G3@SeHANs was 2 % (**New Fig. 2F**), which means there were only 0.2 µg of PAMAM-G3 in 10 µg of G3@SeHANs. Given that the most active component for scavenging cfDNA was PAMAM-G3, our results suggest that G3@SeHANs greatly improved the activity for scavenging cfDNA (by 10 times) compared with the soluble PAMAM-G3. We have added this part into the 2nd Paragraph of the Discussion section.

7. Page 4, Line 111-129 should be presented as the first experiment and finding –aim should be revised.

Reply:

Thank you for the suggestion. We have revised the Results section.

8. Once again, the results section is difficult to follow—confusing statements and not presented in a logical order. Additional subheadings are strongly suggested.

Reply:

Thanks for the suggestion. We have revised the text, and six subheadings for the Results section were added in the revised manuscript.

9. Please clarify what “related phenotypes” was referred to in the following sentence? “The ligature was placed in a circle around the second molar to obtain significant bone loss and related phenotypes”. Also, what does “significant bone loss” refer to? What is significant bone loss? Quantification should be given.

Reply:

The ligature-induced periodontitis model would cause local inflammation-induced gingival mucosa edema and tooth loosening due to alveolar bone resorption. However, among these phenotypes, alveolar bone resorption is the easiest phenotype to quantify by microCT scanning, which is also the most diagnostic pathological change in clinical periodontitis patients. Therefore, we only performed quantitative analysis for alveolar bone loss.

In addition, in patients with periodontitis, diagnosis can be made as long as bone loss occurs. The same diagnostic criteria are generally followed for the evaluation of mouse models. Therefore, the model is well accepted as long as there is a statistically significant difference in the quantitative alveolar bone morphology data (e.g. CEJ-ABC) between the experimental group and the control group. We have replaced the “related phenotypes” with specific descriptions.

As for the “significant” part, we deleted it as there is no certain definition for it, and our intention is to compare the bone loss between each group.

10. The following statements are not clear: PBS, PAMAM-G3, or G3@SeHANs were injected on both sides of mice (what are the locations of the injections?) into six sites around the ligature on days 0, 3, 6, 9, and 12. Samples (what samples?) were collected on day 15. Treatments were also smeared onto the gum (how and why was this performed?) around the ligature daily except on injection days. Please explain the logic and mechanism of topical application and impact on gingiva.

Reply:

Thanks for pointing this out. PBS, PAMAM-G3, or G3@SeHANs were injected on both sides of the mice's second molar of the maxilla, and into six sites around the ligature on days 0, 3, 6, 9, and 12. Maxilla samples were collected on day 15. Treatments were also smeared onto the gum around the ligature daily except on injection days because we wanted to recapitulate the rinsing (Fig. 5A-B). We have clarified these statements in the revised manuscript.

To address the logic and mechanism of topical application, we performed another animal study. Although both cfDNA levels in saliva and serum were correlated with periodontitis, the correlation between the saliva and periodontitis was much higher than that of the serum (New Fig. 1C). Moreover, when applying another NABNs (mesoporous silica nanoparticles functionalized with polyethylenimine, MSN-PEI) targeting circulating cfDNA that have been applied to systemic inflammatory disease like sepsis⁶, we found that circulating cfDNA decreased, but the local bone loss was not alleviated. Moreover, G3@SeHANs were not appropriate for intraperitoneal injection. Therefore, we decided on the topical application, including injecting the materials into periodontal tissue and smearing the materials on the gingiva. The topical application worked well. We have revised the respective part of the Results, Discussion, and Methods.

11. The authors indicate that saliva was collected after anesthesia—which is normally contradictory to saliva secretion, anesthesia cause dry mouth in mice—was saliva secretion induced by any pharmacological agent (pilocarpine, for example)? How much saliva was obtained?

Reply:

This is a very interesting question. Indeed, as stated by the reviewer, we also observed inhibition of salivary secretion during daily anesthesia with isoflurane. However, for the convenience of oral microscopic operation, we chose to use Zotetil 50 (zolazepam+tiletamine) for intramuscular injection anesthesia in this experiment.

It is recommended to inhibit glandular secretion by injecting atropine 15 minutes prior to anesthesia using Zotetil 50. However, we found that salivary secretion was unaffected in mice anesthetized when no atropine was used. Therefore, we chose to collect saliva under this anesthetic condition. It is worth mentioning that the body position of mice matters when using this method of anesthesia, otherwise choking is likely to occur. This might be the reason why pre-anesthetic injection of atropine is recommended. We have included this observation in the new Methods section.

12. Were ligatures kept during the 2 weeks? Were there any loss of ligatures and replacement during this period?

Reply:

The ligatures were kept during the whole 2 weeks. During our preliminary study, loss of ligatures happened a little bit more, mainly due to the untight knot. After practicing many times, it could be solved. Also, we checked the ligatures every day, as we needed to smear the nanoparticles. When we found the loss, we would replace a new ligature. In any event, it happened in no more than 5 mice, out of more than 80 mice we worked on.

13. Why the plaque samples were collected with swaps instead of by ligatures?

Reply:

In fact, the ligature was also collected from mice. But we found that the amount of DNA extracted from the ligature did not meet the sequencing requirements for 16s rDNA. Therefore, we referred to another sample collection method specially for 16s rDNA-seq¹³. We have also added this reference to the Methods section.

14. Maxilla was first used for microCT then for histology, but also for protein extraction. Please clarify, which maxillary samples were used for this.

Reply:

Thanks for pointing this out. The samples for micro-CT and histology and the samples for protein extraction were from the same groups but different mice. For DNA and RNA extraction, half of the maxilla on a random side was dissected and rinsed in cold PBS. The crown of the molars, maxillary bone, and buccal soft tissues were removed from samples, and only the gingival tissues and alveolar bone around the three molars were collected. The trimmed maxillary sample was temporarily placed on dry ice and stored at -80 °C for further testing. Then, for microCT and histology, the other side of the maxilla, heart, liver, spleen, lung, and kidney were dissected and fixed in 4% paraformaldehyde at 4 °C overnight. The fixed samples were rinsed with tap water for 6 h and were stored in 70% ethanol at 4 °C until further testing.

15. 16S RNA shows no impact on microbiome in total, need to add to discussion the explanation of these results.

Reply:

The lack of disruption of microbiota could be due to the injection of materials into tissues; any selenium released from the tissue may not influence the balance of oral surface microbiota. We included the explanation in the new Discussion section.

16. Fig S18—the differences hard to be seen. Please explain.

Reply:

Many thanks to the reviewer for pointing this out. Unfortunately, this was indeed an error in our writing. Significant difference of CD11c+ cells could not be found between groups,

which is also the reason why we did not put labels on the figure. In addition, this is also one of the reasons why we focused on the mononuclear phagocytic system in the subsequent sections. We have stated that "...G3@SeHANs did not affect the number of dendritic cells (DC) in periodontal tissue" in the revised manuscript.

17. "cfDNA and TNF- α levels in saliva and serum were significantly higher in periodontitis model mice than in normal mice; these proinflammatory cytokine levels (there is only one cytokine measured) were inhibited by treatment with..."

Reply:

Thank you, we have corrected the error.

18. Were treatments initiated after 2 weeks of periodontitis induction? What do days 0, 3, 6, 9, and 12 mean?

Reply:

Treatment was initiated when the animal model was established. Please refer to the new Fig.5 A, which demonstrated experimental schedule of in vivo study.

19. MicroCT images (Fig. 3I-J) do not show much bone loss reduction as stated by the authors. Please explain.

Reply:

In the ligature model, bone loss happens rapidly within 15 days. The bone loss is inhibited by 1/4 to 1/3. In the Micro-CT images, the loss at the second molars was significant in that 2/3 of the roots were exposed in the untreated group, clearly more so than in the treated groups.

20. Histological images show similar degree of inflammation, it is hard to see the differences.

Reply:

It can be found in HE staining that the integrity of the epithelium, the width of the alveolar bone and the degree of subepithelial immune cell infiltration was improved in the G3@SeHANs-treated group (New Fig.6A and New Fig. S21A). We detailed it in the respective New Results section.

21. The ligature model used is more of a traumatic model (5-0 sutures) than bacterial inflammatory model.

Reply:

The point is well taken. In fact, there has been an unending discussion on the superiority of the two animal models, namely ligature-induced and periodontal pathogen (bacterial)-induced periodontitis (LIP and BIP).

Comparatively, the BIP model is more mimetic of human chronic periodontitis with respect to similar pathogenic factors and disease progression. However, a long time period is needed for the onset of BIP, and the degree of bone loss is unstable. We have tried a variety of pathogenic bacteria (or combinations) and method of oral bacterial application, but could not obtain consistent and obvious alveolar bone loss. In contrast, the LIP model has a shorter onset period and a more stable and severe bone loss phenotype. We acknowledge that the influence of trauma on the pathogenesis of LIP cannot be excluded. However, a study specifically devoted to the establishment of LIP models showed that the administration of antibiotic treatment to LIP mice significantly alleviated the alveolar bone loss¹³. This suggests that the dysbiotic oral microbiota plays an important role in the pathogenesis of LIP. In addition, since tartar (mineralized plaque, soft scale and food residue around gingival sulcus) is one of the main clinical manifestations and pathogenic factors of periodontitis, the traumatic damage of the local gingival barrier is also one of the driving factors of the progression of periodontitis.

Also, other study on the local pathological immune response in periodontitis have also chosen LIP as an animal model, and the results also show that the periodontitis patients shared similar oral mucosal immunopathology with the LIP model¹⁴. In summary, based on the stability of the model, the significance of the phenotype and at least partial consistency between the LIP model and the periodontitis, we believe LIP was an acceptable animal model used in this paper. Besides, the scavenging of DAMPs was also evaluated in *in vitro* experiments with comprehensive consideration of the traumatic factor. We have included a few text discussing the periodontitis models in the revised manuscript.

22. The relevance of these periodontitis models to human chronic periodontitis should be discussed in the discussion especially those aggressive and acute bone loss models.

Reply:

Thank you again. We have added forementioned part (in the response to Q21) in the discussion section.

23. It is interesting that the most benefit was shown by G3@SeHAN was extracellularly localized, as it is know that cfDNA was more involved in cytoplasm. Please explain.

Reply:

Thanks for the comment. The concept of cfDNA is more often considered as the collection of all the molecules that are outside the cell (cell-free) and contains DNA. This includes: truly free DNA without proteins, and complexes of DNA and proteins (including histone and other DNA binding protein). It is true that such a nomenclature (cfDNA for all these molecules) is not appropriate from a biochemical point of view, since proteins-DNA complexes cannot be referred to as DNA, but unfortunately there is no more appropriate name to describe this

collection of substances. There are different receptors for cfDNA. TLR9 localizes in the endolysosomes, and cGAS is a representative receptor localized in cytoplasm. In our study, we demonstrated the cfDNA-scavenging strategy in inhibiting TLR9 activation. TLR9 preferentially recognizes DNA sequences containing unmethylated cytosine-phosphate-guanosine (CpG) oligodeoxynucleotides, which are highly frequent motifs in bacterial DNA and rare in the mammalian genome, and triggers proinflammatory cytokine response. Cell-free DNA (cfDNA) includes endogenous nuclear and mitochondrial DNA released by damaged host cells, and exogenous bacterial or viral DNA, that can be regarded as the ligand to the TLR9. Thus, G3@SeHAN can scavenged those DNA extracellularly. Also, soluble NABP scavenges cfDNA intracellularly and can also be used to deliver cfDNA into cells in gene delivery. Intracellular scavenged cfDNA could further activate the proinflammatory pathway. Extracellular nucleic acid-binding nanoparticles might avoid this possibility because the cfDNA is scavenged before cell entry, preventing interactions between agonists and TLR receptors.

We have revised the text to provide a clearer description of cfDNA. We have also elaborated it in the Discussion section.

24. English grammar use and writing of entire manuscript (organizational) should be reviewed and reorganized.

Reply:

We have revised the entire manuscript with careful proofreading and polishing. We highlighted the changes in the revised manuscript.

Reviewer #3 (Remarks to the Author):

The manuscript entitled " Cationic Nanostructures for the Treatment of Inflammatory Bone Loss " developed a bone-mimicking 28 selenium-doped hydroxyapatite nanoparticles with cationic polyamidoamine dendrimers 29 (PAMAM-G3), and compared the activities of these NABNs with those of soluble PAMAM-G3 30 polymers for treatment of inflammatory bone loss. Generally speaking, the introduction is insufficient, and the authors should provide more evidence to support the formation of the bone-mimicking nanoparticles. Therefore, I cannot accept it at the current stage:

1. The introduction in the present study does not make the significance and impact of the work clear.

Reply:

We have rewritten the manuscript to clarify our work including the significance and impact. Please see the new Introduction section.

2. What amount of Se was doped into HA? How to determine it?

Reply:

We have detected the amount of Se doped in SeHANs and G3@SeHANs by ICP-OES, and their Se content were $31.902 \pm 1.456 \mu\text{g}/\text{mg}$ and $27.335 \pm 0.615 \mu\text{g}/\text{mg}$, respectively (New Fig. S9 B). We also determined that the valence state of Se in SeHANs was Se(IV), which was involved in selenite groups (New Fig. 2D).

3. The authors should carefully clarify why they synthesize nanoparticles are bone-mimicking in terms of chemistry, structure or biological property. Because the apatite in bone is complicated and has different configurations, it is recommended to compare their synthesized "bone-mimicking" nanoparticles with natural apatite nanoparticles in bone, so that to demonstrate that their fabricated nanoparticle are bone-mimicking.

Reply:

We claim that the synthetic selenium-doped HA nanoparticles (SeHANs) are **bone-mimicking mainly from the morphological and structural perspective**. The natural bone apatite nanocrystals are uniaxially oriented align with respect to the collagen fibril axis. Our synthetic SeHANs are a highly ordered and uniaxially oriented hierarchical structure alike to natural bone apatite, despite no collagen is used as a macromolecular template (New Fig. 2A)¹⁵. Further, when compared with Bio-Oss[®], which is considered to be collagen-free natural bone apatite¹⁶, **the synthetic SeHANs are similar to natural bone apatite nanoparticles in both single crystal and hierarchical structure**. Please see the **New Fig. S5**.

From a chemical standpoint, the natural bone apatite often has other elements substitution besides Ca^{2+} , PO_4^{3-} and OH^- , typically carbonate. We therefore compare the FTIR spectra of synthetic SeHANs with natural bone apatite (Bio-Oss[®]) and the result is shown in **New Fig. S6**.

The typical bands of PO_4^{3-} ($1200\text{-}900$, 603 , 562 cm^{-1}) and OH^- (3570 cm^{-1}) are present in both of SeHANs and natural bone apatite. More importantly, carbonate bands at 872 cm^{-1} and in the range of $1600\text{-}1400 \text{ cm}^{-1}$ are also found in both of natural bone apatite and synthetic SeHANs. **These results demonstrate a high similarity of chemical composition between SeHANs and natural bone apatite**. It may be noticed that in natural bone apatite, there are bands belonging to H_2O at $3750\text{-}3000 \text{ cm}^{-1}$ and 1633 cm^{-1} , which are not obvious in SeHANs. And the bands of the asymmetric stretching vibration of CH_2 groups at 2962 and 2854 cm^{-1} are exclusively present in the synthetic SeHANs. These differences are due to the lack of water and the high content of long-chain fatty acids (linoleic acid) and long-chain fatty amines (octadecylamine) in the LSS synthetic system of SeHANs. However, the existence of these differences cannot repudiate the fact that SeHANs mimic the characteristics of multi-element substitution in natural bone apatite.

From the aspect of phase characteristics, the element substitution in natural bone apatite leads to the decrease of crystallinity¹⁷. Therefore, we compared the XRD patterns of SeHANs with natural bone apatite (Bio-Oss[®]). Please see New Fig. S7. The typical peaks of hydroxyapatite were present in both synthetic SeHANs and natural bone apatite and no obvious secondary phase was found in both of them. Noticeably, the diffraction peaks of SeHANs were more broadened than natural bone apatite, which not only indicates that SeHANs mimic the low crystallinity of natural bone apatite but also demonstrates that selenium doping further decrease the crystallinity of SeHANs in comparison with natural bone apatite.

As for biological properties, the synthetic SeHANs not only showed good biocompatibility and promoting effects in osteogenic differentiation of MSCs, but also induced autophagy and apoptosis in osteosarcoma cells for bone tumor inhibition^{15,18}.

We have included the clarification on the bone-mimicking of SeHANs in the Results and Discussion sections of the revised Manuscript.

4. The pictures should be further improved, and the pictures should be aligned with each other. For example, Figure 1a and Figure 1b are not aligned.

Reply:

The point is well taken. We have rearranged the figures to make them more attractive.

5. Between each picture, the font format and symbol format should be consistent within the same picture. Please adjust them.

Reply:

Thank you for the suggestion. We have redrawn the figures accordingly.

References:

- 1 Holdenrieder, S. *et al.* Nucleosomes in serum as a marker for cell death. *Clin Chem Lab Med* **39**, 596-605, doi:10.1515/cclm.2001.095 (2001).
- 2 Abrams, S. T. *et al.* Circulating histones are mediators of trauma-associated lung injury. *Am J Respir Crit Care Med* **187**, 160-169, doi:10.1164/rccm.201206-1037OC (2013).
- 3 Nakahara, M. *et al.* Recombinant thrombomodulin protects mice against histone-induced lethal thromboembolism. *PLoS One* **8**, e75961, doi:10.1371/journal.pone.0075961 (2013).
- 4 Xu, J., Zhang, X., Monestier, M., Esmon, N. L. & Esmon, C. T. Extracellular histones are mediators of death through TLR2 and TLR4 in mouse fatal liver injury. *J Immunol* **187**, 2626-2631, doi:10.4049/jimmunol.1003930 (2011).

- 5 Liu, F. *et al.* A Cationic Metal-Organic Framework to Scavenge Cell-Free DNA for Severe Sepsis Management. *Nano Lett* **21**, 2461-2469, doi:10.1021/acs.nanolett.0c04759 (2021).
- 6 Dawulieti, J. *et al.* Treatment of severe sepsis with nanoparticulate cell-free DNA scavengers. *Sci Adv* **6**, eaay7148, doi:10.1126/sciadv.aay7148 (2020).
- 7 Liang, H. *et al.* Cationic nanoparticle as an inhibitor of cell-free DNA-induced inflammation. *Nat Commun* **9**, 4291, doi:10.1038/s41467-018-06603-5 (2018).
- 8 Nishimoto, S. *et al.* Obesity-induced DNA released from adipocytes stimulates chronic adipose tissue inflammation and insulin resistance. *Sci Adv* **2**, e1501332, doi:10.1126/sciadv.1501332 (2016).
- 9 Snyder, M. W., Kircher, M., Hill, A. J., Daza, R. M. & Shendure, J. Cell-free DNA Comprises an In Vivo Nucleosome Footprint that Informs Its Tissues-Of-Origin. *Cell* **164**, 57-68, doi:10.1016/j.cell.2015.11.050 (2016).
- 10 Marsman, G., Zeerleder, S. & Luken, B. M. Extracellular histones, cell-free DNA, or nucleosomes: differences in immunostimulation. *Cell Death Dis* **7**, e2518, doi:10.1038/cddis.2016.410 (2016).
- 11 Barton, G. M., Kagan, J. C. & Medzhitov, R. Intracellular localization of Toll-like receptor 9 prevents recognition of self DNA but facilitates access to viral DNA. *Nat Immunol* **7**, 49-56, doi:10.1038/ni1280 (2006).
- 12 Kim, P. D. *et al.* Toll-Like Receptor 9-Mediated Inflammation Triggers Alveolar Bone Loss in Experimental Murine Periodontitis. *Infect Immun* **83**, 2992-3002, doi:10.1128/iai.00424-15 (2015).
- 13 Abusleme, L. *et al.* Oral Microbiome Characterization in Murine Models. *Bio Protoc* **7**, doi:10.21769/BioProtoc.2655 (2017).
- 14 Dutzan, N. *et al.* A dysbiotic microbiome triggers T(H)17 cells to mediate oral mucosal immunopathology in mice and humans. *Sci Transl Med* **10**, doi:10.1126/scitranslmed.aat0797 (2018).
- 15 Li, X. *et al.* Hierarchically constructed selenium-doped bone-mimetic nanoparticles promote ROS-mediated autophagy and apoptosis for bone tumor inhibition. *Biomaterials* **257**, 120253, doi:10.1016/j.biomaterials.2020.120253 (2020).
- 16 Orr, T. E., Villars, P. A., Mitchell, S. L., Hsu, H. P. & Spector, M. Compressive properties of cancellous bone defects in a rabbit model treated with particles of natural bone mineral and synthetic hydroxyapatite. *Biomaterials* **22**, 1953-1959, doi:10.1016/s0142-9612(00)00370-7 (2001).
- 17 Ma, J., Wang, Y., Zhou, L. & Zhang, S. Preparation and characterization of selenite substituted hydroxyapatite. *Materials Science and Engineering: C* **33**, 440-445, doi:<https://doi.org/10.1016/j.msec.2012.09.011> (2013).
- 18 Li, Y. *et al.* Assembly Mechanism of Highly Crystalline Selenium-Doped Hydroxyapatite Nanorods via Particle Attachment and Their Effect on the Fate of Stem Cells. *ACS Biomaterials Science & Engineering* **5**, 6703-6714, doi:10.1021/acsbomaterials.9b01029 (2019).

Reviewers' Comments:

Reviewer #1:

Remarks to the Author:

Nanoparticulate Cell-free DNA Scavenger for Treating Inflammatory Bone Loss in Periodontitis

In the revised manuscript, the aim of the study is much clear, and many of the technical deficiencies have been resolved and clarified.

With regard to the experiment to show the presence of chromatin particles or protein-bound DNA and free DNA, the authors can perform immunofluorescence. They can label histones and DNA with specific fluorescently tagged antibodies against respective molecules and then observe the single labeled and dual labeled molecules. The molecules with co-localization of both fluorescences will be chromatin. This experiment will be worth performing to understand the nature of circulating DNA particles contributing to periodontitis.

Reviewer #2:

Remarks to the Author:

Thank you for the authors for their detailed responses and revisions made in the resubmitted paper. There are still areas of grammatical errors, however I trust that those could be further corrected.

The main concern at this time is that the authors did not mention or discuss the potential method of applying these nanoparticles in human periodontal treatment- in mice the nanoparticles were injected at six sites of the tooth at every 3rd day and also topically given to the gingival margins. How would these applications be translated to clinic? Injections at 6 sites of each tooth at certain intervals may not be practical. Would topical application be as effective as injection?

Also, the authors mentioned that the reason injection were given at every 3rd day was to avoid gingival irritation and toxicity. Any observation can be shared in animal model? From histological section, a damage to interdental gingival epithelium could be seen. Can this be due to injections or due to ligature?

Similarly, the safety and tolerability in human clinical applications should be discussed.

Response to Reviewers' comments

We would like to thank the reviewers and editors for their constructive comments.

We have complied with all the editorial requests.

Editorial policy checklist, reporting summary, and 'Source Data' file have been updated.

Please find our detailed response in a point-to-point manner as listed below.

Corresponding changes have been made in the manuscript and Supplementary information as indicated.

Comments from Editor:

In particular, we ask that a revised manuscript convincingly addresses the concerns of reviewer 2 on the clinical applicability of the NPs, and those of reviewer 1 on distinguishing histone-bound from cell-free DNA. This is not to say that we consider the other concerns of our reviewers any less important. We evaluated your responses to reviewer 3 in-house. We ask that you reconsider using the terminology 'bone-like', as it is not essential to discuss the advance but, as in the case for reviewer 3, readers might also disagree with the definition.

Reply:

Thanks for your kind suggestions. We have carefully revised our manuscript according to the comments by reviewers 1 and 2, and also included new data to address reviewer 1's request to distinguish histone-bound DNA from free DNA; we explained why we could not finish the experiment proposed by reviewer 1 due to the technical limitation of immunofluorescence. Regarding the suggestion by reviewer 3, we have abandoned all the terminologies or discussions related to "bone-like" or "bone-mimicking".

Reviewer #1 (Remarks to the Author):

In the revised manuscript, the aim of the study is much clear, and many of the technical deficiencies have been resolved and clarified.

With regard to the experiment to show the presence of chromatin particles or protein-bound DNA and free DNA, the authors can perform immunofluorescence. They can label histones and DNA with specific fluorescently tagged antibodies against respective molecules and then observe the single labeled and dual labeled molecules. The molecules with co-localization of both fluorescences will be chromatin. This experiment will be worth performing to understand the nature of circulating DNA particles contributing to periodontitis.

Reply:

We thank the reviewer for the suggestion to explore the nature of circulating cfDNA. We tried to perform immunofluorescence imaging of chromatin in serum as suggested by the reviewer. We used histone H3 antibody in the pull-down experiment and anti-ds DNA antibody (Abcam, ab27156) for antigen recognition.

Unfortunately, we were unable to separate the antigen-antibody complex from the unbound antibody after the primary antibody incubation step. In conventional immunofluorescence staining, the unbound antibody must be removed; but in our case, we could not wash out the antibody because of the DNA presence in the fluid. We then searched the literature to adopt a protocol that could detect the free antigen in solution by performing co-localization immunofluorescence imaging. Unfortunately, that was also unsuccessful. It is noteworthy that the diameter of the nucleosome is around 11 nm, which is below the theoretical optimal resolution of conventional confocal microscopes (180 nm-500 nm). We therefore could not acquire the desired fluorescence images of chromatin and free-DNA.

Fortunately, we obtained data that might help address the concerns to some extent. After submission of the revised manuscript, we continued to optimize the protocol for the histone H3 pull-down experiment, and acquired the following results.

In brief, we previously hypothesized that the high concentration of albumin in the serum would affect the recognition of histone H3 by the primary antibody, while high concentrations of serum IgG would affect the capture of the adding primary antibody by the magnetic beads. In subsequent attempts, we used protein A/G magnetic beads instead of protein A sefinose to pre-capture the natural IgG in the serum. Then, instead of using the albumin affinity gel, we reduced the concentrations of non-target proteins by diluting the sample eight-fold with PBS. We used the fragment size characteristics of cfDNA to estimate its origin.

Most of the DNA fragments detected on the beads were below 200bp in size, exhibiting a broad distribution. Around 150 bp, the classical size of DNA within a single nucleosome, a small distribution peak could be observed in both samples. This suggests that chromatin might have been captured on the beads. The atypical distribution pattern of DNA size may be a result of cfDNA degradation caused by the process of pull-down and DNA extraction from the beads. In addition, in the corresponding post-pull-down serum samples, we detected a lower concentration of DNA with size characteristics above 1000 bp, which are more consistent with those of saliva-derived cfDNA (Fig. S13) (Fig. 1 in this text).

We will be very cautious in our interpretation of the results above. First, based on the DNA detected on beads and its size distribution pattern, it is very likely that a portion of the serum-derived cfDNA is present in the form of chromatin (DNA-histone complex). DNA with different distribution patterns from chromatin could still be detected in the samples after the H3 pull-down, suggesting that a smaller amount of cfDNA may be present in the form of histone-free DNA. However, due to the lack of means to accurately quantify the capture efficiency of histone H3 and the potential attrition and degradation during the experiment, the different cfDNA concentrations in the two fractions may not reflect the true ratio of cell-free DNA-histone complex and histone-free DNA in serum. Nevertheless, it would be reasonable to infer a higher proportion of cell-free DNA-histone complex than histone-free DNA in serum cfDNA.

We have combined the above results on cfDNA size distribution in saliva and serum to form the revised Fig. S13 (Fig. 1 in this text), and included new text to discuss how circulating DNA particles contribute to periodontitis in the revised manuscript. The detailed experimental procedures have also been added to the methodology section. We hope that these results inferring the nature of circulating DNA can partially satisfy the critique.

Fig. 1. The size distribution of cfDNA fragments in saliva and serum and serum cfDNA detected after histone H3 pull-down (Fig. S13 in the revised manuscript)

(A) The cfDNA was extracted from the serum and saliva of two healthy participants. The same DNA samples were used for DNA concentration determination and detection of DNA fragment size distribution, respectively. Serum cfDNA samples were diluted twice before analysis. The red arrows indicate the characteristic peaks in the size distribution of serum cfDNA. (B) Serum samples from another two healthy participants were collected for histone H3 pull-down. The left column shows the size distribution of cfDNA extracted from the remaining serum sample after histone H3 pull-down; the right column shows the size distribution of DNA extracted from the magnetic beads (captured by histone H3 antibody). The blue arrows indicate the small peaks around 150bp.

Reviewer #2 (Remarks to the Author):

1. There are still areas of grammatical errors, however I trust that those could be further corrected.

Reply:

Thank you for the alert. We again carefully proofread the manuscript and corrected all the grammatical errors.

2. The main concern at this time is that the authors did not mention or discuss the potential method of applying these nanoparticles in human periodontal treatment- in mice the nanoparticles were injected at six sites of the tooth at every 3rd day and also topically given to the gingival margins. How would these applications be translated to clinic? Injections at 6 sites of each tooth at certain intervals may not be practical. Would topical application be as effective as injection?

We appreciate the comment. Dental plaque and tartar in the gingival sulcus initiated periodontitis. Controlling the local inflammatory stimulus in the gingival sulcus would help alleviate the condition. Based on this concept, we first tested whether we could apply the nanoparticles topically in the gingival sulcus of mice when we started our animal study. However, the space was too limited to inject the nanoparticles. We then decided to inject the nanoparticles at six sites around the ligatured tooth, and smeared the nanoparticles on the gingival margin. This study is a proof-of-concept of the cfDNA-scavenging approach to treating periodontitis. In patients, topical application of these nano-scavengers is certainly feasible. There are already many topical formulations for gingival sulcus. For example, after dental scaling, subgingival minocycline hydrochloride ointment would be injected into the gingival sulcus for controlling the local inflammation.¹ In our case, the nanoparticles can be readily mixed with a hydrogel for application to the gingival sulcus. Meanwhile, there are also other cfDNA-scavenging materials that can be applied to clinical periodontitis treatment as hydrogels.²

We have included new text discussing the possibility of translating the cfDNA-scavenging approach for treating periodontitis in the revised manuscript.

3. Also, the authors mentioned that the reason injection were given at every 3rd day was to avoid gingival irritation and toxicity. Any observation can be shared in animal model? From histological section, a damage to interdental gingival epithelium could be seen. Can this be due to injections or due to ligature?

Thanks for your comment. In our pilot study, injection every other day would cause swollen gingival tissue in some mice. In addition, the anesthesia given to the animals would weaken the mice if the injection was too frequent. Besides the reason motioned in

response to comment 2 and the aforementioned reason, we decided to give the injection once every three days and also topically smeared the nanoparticles on the gingival margins. We have included a new text to explain the rationale of the experimental design in the revised manuscript.

To answer whether the damage to the interdental gingival epithelium was caused by injections or ligatures, we would like to show some images from another project. We observed the effect of periodontal microinjection on alveolar bone resorption and gingival tissue destruction. The results showed that if the microinjections were performed at the same frequency and interval as in our manuscript (without LIP model establishment, named as "Sham" group), no alveolar bone resorption and epithelial destruction would be observed by microCT scanning and HE staining (Fig. 2). Thus, the ligature should be the reason for the damage. We would like to report this observation in our future manuscript.

Fig. 2 The effect of periodontal microinjection on alveolar bone resorption and gingival tissue destruction

A. Micro-CT scanning and 3D reconstruction of the alveolar bone. (scale bars: 500µm)

B. Bone loss measured by the vertical distance between the cemento enamel junction (CEJ) and alveolar bone crest (ABC) (CEJ-ABC). Data are means \pm SD; ns: not significant by unpaired Student's t test.

C. H&E staining of periodontal tissues. (scale bars: 500µm)

4. Similarly, the safety and tolerability in human clinical applications should be discussed.

Thank you for the suggestion. As we mentioned above, we can apply the nanomaterials topically to the gingival sulcus in the human application, which is a noninvasive maneuver that would cause minimal damage to the tissue. Meanwhile, hydrogel

formulation could prolong the retention of the nanomaterials in the gingival sulcus and improve safety at the same time. As for tolerability, there are already many drugs applied in a topical manner to the gingival sulcus. For example, Compound Chamomile and Lidocaine Hydrochloride Gel are used three times a day in periodontitis patients' gingival sulcus for controlling inflammation in our hospital. Minocycline hydrochloride ointment is also used once a week in periodontitis patients' gingival sulcus. Thus, topical application in the gingival sulcus is tolerable.¹ We have included a few texts to discuss the promise of clinic translation in the revised manuscript.

References:

- 1 van Steenberghe, D. *et al.* Subgingival minocycline hydrochloride ointment in moderate to severe chronic adult periodontitis: a randomized, double-blind, vehicle-controlled, multicenter study. *J Periodontol* **64**, 637-644, doi:10.1902/jop.1993.64.7.637 (1993).
- 2 Yang, C. *et al.* An Injectable Antibiotic Hydrogel that Scavenges Proinflammatory Factors for the Treatment of Severe Abdominal Trauma. *Advanced Functional Materials* **n/a**, 2111698, doi:<https://doi.org/10.1002/adfm.202111698>.

Reviewers' Comments:

Reviewer #1:

Remarks to the Author:

The revised version of the manuscript is more descriptive with also focus on the clinical application of this study. The authors' clarification on cell-free DNA including both free DNA as well as histone-bound DNA is acceptable.

Response to Reviewers' comments

We would like to thank the reviewers and editors for their constructive comments.

Comments:

The revised version of the manuscript is more descriptive with also focus on the clinical application of this study. The authors' clarification on cell-free DNA including both free DNA as well as histone-bound DNA is acceptable.

Reply: Thank you so much for your constructive suggestions during our submission, which indeed improve our manuscript. We really appreciate it.